# A Travel through Landscapes of Seed Dormancy

**DOI:** 10.3390/plants12233963

**Published:** 2023-11-24

**Authors:** Alberto Gianinetti

**Affiliations:** Council for Agricultural Research and Economics (CREA)—Research Centre for Genomics and Bioinformatics, Via S. Protaso 302, 29017 Fiorenzuola d’Arda, PC, Italy; albgian@libero.it

**Keywords:** seed dormancy, dormancy cycling, dormancy continuum, functional model, abscisic acid, bistable regulatory circuit, tristable regulatory circuit, development landscape

## Abstract

Basic features of seed dormancy are illustrated. The seed overall regulatory network governs seed metabolism and development, and it is coordinated by plant hormones. A functional model focused on abscisic acid (ABA), the foremost plant hormone in dormancy, is used as a framework to critically discuss the literature. Gibberellins (GAs) have a main role in germination, and the ABA–GAs balance is a typical feature of the seed state: ABA dominates during dormancy and GAs prevail through germination. Thus, the literature converges toward envisaging the development switch between dormancy and germination as represented by the ABA/GAs ratio. The ABA–GAs antagonism is based on mutual inhibition, a feature of the regulatory network architecture that characterizes development trajectories based on a regulatory circuit with a bistable switch. Properties of such kind of regulatory architecture are introduced step by step, and it is shown that seed development—toward either dormancy or germination—is more properly represented by a tristable regulatory circuit, whose intermediate metastable states ultimately take one or the other development trajectory. Although the ABA/GAs ratio can conveniently represent the state of the seed overall regulatory network along the seed development trajectory, specific (unknown) dormancy factors are required to determine the development trajectory. The development landscape is shown to provide a well-suited representation of seed states travelling along developmental trajectories, particularly when the states are envisioned as regulatory circuits. Looking at seed dormancy in terms of regulatory circuits and development landscapes offers a valuable perspective to improve our understanding of this biological phenomenon.

## 1. Basic Features of Seed Physiological Dormancy

Dormancy is the eco-developmental arrest of a meristematic or embryonic organ, whereby growth fails to respond to favourable conditions until sufficient entrainment by environmental cues occurs [1]. Often, rather than a complete arrest of growth, dormancy manifests as slow development, even under optimal conditions, as compared to the speed of development that occurs after dormancy has been removed. For example, in bulbs, dormancy frequently shows up as delayed growth, so that the criterion used to define its complete removal is, rather than bare sprouting or plantlet emergence, the development of a bulb bud into a rapidly elongating sprout, which then grows into a normal plant [2]. What, exactly, “rapidly elongating” means must be defined empirically, but presently, this is the best, ex post, criterion to assess full dormancy release. Although dormancy features are common among buds, bulbs, seeds, and other meristematic or embryonic organs [1,3], in this review, I will focus on seed dormancy.

In dispersal units (seeds or caryopses or more complex structures), dormancy is assessed only indirectly, as a lack, or reduction, of germination by living seeds (where ‘seeds’ is used in a broad sense through this review) that are imbibed under conditions otherwise favourable for germination [4,5,6]. Germination is the process by which a seed develops into a seedling, and, sensu stricto, this transition is completed by the emergence of the embryo (the radicle, in most species) from its surrounding tissues [5]. This visible event is traditionally designated as ‘germination’, but it is, in fact, a snapshot taken during a continuous growth process [7]. Thus, if a viable seed does not germinate in suitable conditions, it is inferred to be dormant since it shows no sign of development or growth. A seed sample is said to be partially dormant if its germination is lower, or slower, than that of a sample of non-dormant seeds. In general, the necessity of defining dormancy in terms of reduced, or slower, germination/sprouting with respect to non-dormant seeds/buds, creates a definition loop, especially when one needs to establish whether dormancy has been fully removed. Trial-and-error was the historical way to (understand and) solve this problem; experience with a specific plant material is the practical expedient used thereafter.

Germination is classically divided in three phases based on the dynamics of seed water uptake [5]. The first phase (phase I) is characterized by rapid seed imbibition, and the metabolism of the quiescent dry seed is gradually reactivated. During the second phase (phase II), water uptake slows down, and the metabolism of the hydrated seed is fully activated. In the third phase (phase III), rapid water uptake resumes because of seedling growth. A dormant seed remains in the second phase with a stable water content (unless the soil in which it is located re-dries). This temporal division based on the seed water relations is simplistic, but undisputed.

Seed dormancy regards mature seeds that have dehydrated (at least in orthodox seeds, that is, seeds that are physiologically predisposed to drying), and dormancy is slowly removed when these seeds undergo conditions that break it [6]. In many species, seed desiccation has the important effect of triggering the commutation the developing programme of the seed from embryonal to germinative [8]. Lack of dormancy in the mature seed is, therefore, a phenomenon distinct from pre-harvest sprouting, which can occur in non-dormant species such as maize [9], under warm and wet conditions. Of course, dormancy strongly helps preventing this related phenomenon in species with dormant seeds.

Seed dormancy is an evolutionary adaptation particularly important in seasonally cold and/or drought-prone ecosystems, whereas lack of dormancy is more common in constantly warm and wet climates [10]. There is great diversity in kinds of seed dormancy [11], but physiological dormancy is the most common class of dormancy and is found in many species across all vegetation zones on Earth [6].

Physiological dormancy is induced during seed maturation and can be relieved by afterripening the dry seeds (that is, by storing them at warm temperatures) or via stratification of imbibed seeds [5], where stratification means to expose the imbibed seeds to either warm (>15 °C) or cold (0–10 °C) temperatures, depending on the species [6]. In nature, cold stratification (aka moist chilling) is the commonest way to break physiological dormancy across taxa [6], and it is typical of summer annuals, in which dormancy is broken during the wet and cold winter months. In this way, seeds germinate in the spring or early summer and the plant produces new seeds by the autumn of the same year. In many summer annuals, dormancy is effectively removed by dry afterripening too.

The dormancy-relieving capability of dry afterripening is well known in winter annuals across many plant families, like grasses and many dicots, such as arabidopsis ecotype ‘Cape Verde Islands’. In winter annuals, seeds come out of dormancy during summer, due to high temperatures and dry conditions (that is, because of dry afterripening), and they germinate when it is relatively cool and humid in autumn [6,12,13]. Cold stratification may be effective in these species as well, but to complicate this matter, long expositions to moist chilling can re-induce [13], or even induce [14], dormancy in seeds of some species. In red rice, incubation at a temperature suboptimal for germination (15 °C) induces dormancy, whereas cold stratification (1 °C) promotes germination when the seeds are subsequently incubated at optimal temperature (30 °C), indicating the existence of a temperature response threshold [15]. Germination and induction of dormancy can, indeed, be concurrent processes, leading to opposite fates in seeds of the same genotype under the same incubation conditions [15,16,17].

Arabidopsis and most grasses, if dormant, typically show non-deep physiological dormancy; that is, the embryo of a dormant seed, if excised, quickly grows into a normal seedling [6]. Deep physiological dormancy, which is less frequent, occurs when seeds require a long period of cold stratification to come out of dormancy (typically, 3–4 months) and the latter cannot be broken by GAs and persists even if the embryo is excised from the fresh seed [11,18]. It is also known as ‘embryo dormancy’ [6].

The ecological effect of seed dormancy (which is specifically called ‘primary dormancy’ if the seeds were already dormant at dispersal time) is that germination is prevented at a time of the year when the environmental conditions are permissive for germination (Figure 1), but the climate does not support such conditions to remain favourable long enough for seedlings to become established and the plant population to survive and reproduce successfully [4,6,19].

Noteworthy, if seeds whose dormancy has been relieved are subject to environmental conditions that are not fully suited for germination (because, for example, they require light but are buried in the soil), they may re-enter dormancy. This phenomenon is known as secondary dormancy [20]. Dormancy is, therefore, a reversible block to germination with variable intensity [4,12]. Seeds that accrue in the soil, the so-called soil seedbank (Figure 1), can, thus, undergo seasonal fluctuations in dormancy intensity, through an annual dormancy cycle [4,6].

In species with dormant seeds, field germination takes place over a species-specific, or even ecotype-specific [19], seasonal window [4,6] after dormancy is relieved, at least partially (lowest pane in Figure 1), and the season is favourable for germination (middle pane in Figure 1). In temperate regions, temperature and moisture are the main seasonal variables controlling seed dormancy. However, a complex interaction between environmental variables and seed germination capability determines the cyclical change in the soil seedbank through a continuum gradation of dormancy intensities, potentially from full germination to full dormancy [4,6].

Variation in dormancy degree is continuous because the seeds’ physiological responses are slightly different among individual seeds of the same population, even if they are genetically uniform (i.e., isogenic). Thus, the response of a seed population is probabilistically distributed around a population norm. This explains why some grade of partial dormancy is typically observed even in seeds harvested from the same plant [21,22,23].

Seed dormancy, indeed, is not an all-or-nothing phenomenon, but instead, it is tuned so that part of the seeds will germinate at the appropriate time of the season, while others will remain dormant and replenish the soil seedbank (Figure 1), as an insurance against adverse events that could endanger the survival of the population [24]. In addition to seasonal cold and/or dry periods in natural ecosystems, this holds true for many weeds that infest agricultural systems and must periodically cope with tillage and, of course, chemical or mechanical weeding [25]. The seedbank is, indeed, the primary source of weed infestations in crop fields [25].

It is worth remarking that ‘partial dormancy’ is used with two meanings: it can either mean that, in a seed sample, part of the seeds germinate while the others rest in a dormant state, or it can mean that the seeds do germinate but they germinate more slowly than if they were non-dormant. Although having two meanings can be a semantic quandary, the two phenomena are, indeed, two facets of the same physiological condition [22,26]. As each seed can either germinate or not, partial dormancy is a concept that necessarily applies to a seed population when it is specifically meant to indicate that only part of the seeds germinates. If, instead, it is meant to express the idea that some seeds germinate more slowly than if they were fully non-dormant, the concept of partial dormancy may even be applied to an individual seed, but this works only if one already knows how fast a fully non-dormant seed typically germinates. Because of this caveat and differences between seeds, dormancy in general, and partial dormancy in particular, ought to be assessed on a seed sample large enough, rather than on a single seed [27].

Moreover, in some species, after dormancy has been relieved, the seed requires specific environmental conditions to elicit germination, in addition to favourable moisture and temperature conditions [4]. This means that different environmental inputs are required for dormancy breaking and germination triggering [4,6]. For example, arabidopsis further requires light to germinate, even though dormancy has already been broken [12]. Despite this light requirement sometimes having been considered as a final step of dormancy breaking [19], the fact that illumination is almost immediately effective in triggering germination of an imbibed, fully afterripened seed suggests that it is, rather, a germination requirement [6]. Nitrate plays a similar role, as a low concentration of nitrate (around 0.1 mM) in the germination medium is able to promote seed germination in several species, mostly light-sensitive (i.e., positively photoblastic) weeds that show synergistic light and nitrate effects [28]. This eco-physiological fine-tuning of germination is part of the fitting of a species to its ecological niche [6,19,22].

The phenomenon of dormancy extends beyond the boundaries of plant biology [29], so that though many aspects of dormancy regulation are widely different among species, or even within species, due to the evolutionary fitting of the seed physiology to the ecological features and challenges of the diverse environments [6], some common, basic traits are well preserved across the plant kingdom. Thus, comparing findings obtained from studies on dormancy in different species, such as rice (a monocot) and arabidopsis (a dicot), can provide clues about general features of the mechanisms underpinning seed dormancy. In this respect, bud dormancy should be considered too, since it shares a common mechanism with seed dormancy [1,3,30]. This assumes that a conserved core mechanism is common across species and accessions, even though regulatory networks can largely diverge among them [31,32] as each species befits its environmental niche [33,34,35,36]. Native species, indeed, are often exquisitely attuned to their environments [22], and genes tuning seed dormancy and germination are under strong selection in natural plant populations [21].

Changes in regulatory networks play, indeed, a major role in evolutive adaptation, and species-specific differences in the architecture of gene regulatory networks are, therefore, the major determinants of the phenotypic variations observed across organisms [37]. Thus, regulative loci that appear of great importance for regulating dormancy in one species (for example, arabidopsis) do not have close orthologs with the same relevance in taxonomically distant species (e.g., rice). The transcriptional regulator DELAY OF GERMINATION 1 (DOG1), for example, transduces environmental effects during maturation to set the initial depth of dormancy, but it is not further involved in the seasonal changes during dormancy cycling [38]. This suggests an important role for DOG1 as a temperature sensor [39]. It is, therefore, a key mediator between environment clues and the endogenous setting of primary dormancy, not a dormancy factor itself. Nor does it have an ortholog with a so clear-cut function in rice. If one wishes to study the general features of the core mechanism underpinning seed dormancy, the conserved functions, rather than the species-specific regulators, should thus be paid the most attention.

## 2. The Seed Fate as a Binary Event: The Overall Regulatory Network, Trajectories, and Protocols

Under conditions favourable for germination (Figure 1), the regulatory network of a living seed embodies a bifurcating system with only two possible outcomes: germination and dormancy [40]. That is, at the individual level, the seed fate is a binary event [27]. As dormancy is a reversible state, the flip to or from dormancy requires a germination-controlling switch [40]. Thus, though seeds go through a dormancy continuum [6], so that dormancy is a quantitative rather than categorical trait [4,26]; once imbibed, each individual seed can only follow either one of two developmental routes, toward germination or dormancy. Likewise to what occurs for single cells [41,42], each seed’s developmental route corresponds to a distinct dynamic trajectory of the regulatory network [40]. In accordance with this, it has been highlighted that germination competence is determined by the current transcriptomic state of the seed [43].

Although gene expression is subject to stochastic fluctuations among cells and, even more, between alleles of the same cell, robust and reproducible development trajectories are achieved at the tissue level in part because of spatiotemporal averaging [44,45]. At a high-ranking level of regulation, compensation mechanisms must also be present to dampen random fluctuations in the expression of individual genes and to avoid turning such huge molecular variation into equivalent phenotypic variation [46].

The seed, indeed, is composed of cells relatively uniform within a few tissues, whose states overall connote the seed’s own state, as the latter constrains the former [46,47]. Thus, each seed state corresponds to one or the other specific combination of states of the regulatory networks of its constitutive tissues, depending on whether the seed germinates or not. Here, I denote the ensemble of seed tissue regulatory states as the seed “overall regulatory network”. Which trajectory—toward either germination or dormancy—an overall regulatory network will follow is decided during a short window in the first hours of imbibition [39], and this constitutes the master switch of the seed overall regulatory network.

Differently from the fate of individual cells, which undergo stable differentiation into a given tissue type, seeds that undertake the germination trajectory do not settle in a stable state, but progress through development in a sequence of quasi-steady states. The emergence of the radicle (or of the shoot, in soaked rice seeds) from the covering structures conventionally marks the germination end and the start of seedling growth, which is a post-germinative phase [5]. Hence, it is worth noting that only seeds that have not entered the third of the three phases of germination are in a germinative state comparable (albeit transitorily) to the state of dormant seeds imbibed for the same time under the same conditions [5]. Otherwise, the comparison of the metabolism and the overall regulatory network of dormant seeds with those of non-dormant ones is grossly spurious, as differences along the germination time-course are increasingly due to the progress of development in the germinating seeds rather than to the original diversity between dormant and non-dormant ones.

Once attained, development states and trajectories are self-stabilizing, i.e., ‘canalized’ [48], and, therefore, robust to small environmental perturbations [41,48,49]. Fluctuations in the regulatory network are buffered, and, after some commitment point, the gene expression pattern associated with a state is, indeed, maintained even after the original stimulus that set up the corresponding trajectory has disappeared [41,48,49]. Reversal of conditions may, nonetheless, reverse a trajectory, if it occurs before development becomes irreversible [35,41]. In the case of seeds, this is possible until phase II, under conditions unfavourable for germination. This is, indeed, the foundation of dormancy cycling.

The most important feature of the architecture of the regulatory network is that it imposes constraints onto the collective behaviour of genes; that is, individual genes cannot alter their expression independently [41]. Regulatory networks, indeed, largely consist of interconnected functional modules working within a hierarchical structure [37], and the regulatory state at a higher level of a system constrains the states at a lower hierarchical level [46,47]. At every development state, cellular mechanisms ensure the stability of an organism’s phenotype because of homeostatic settings, that is, sets of executing rules that co-ordinate a biological system to ensure a robust performance [50]. These sets of rules exist as ‘protocols’, that is, informational entities hierarchically higher than genes and even than multigene complexes encoding regulated biochemical pathways [50,51].

Protocols are emerging properties of the biological systems, inherent to the architecture of the regulatory network and “hard-wired” in the genome [41,50,51]. They act as built-in constraints (or boundary conditions), evolved to guarantee robustness of metabolism and phenotype within each development state [51]. Although every regulatory network has broad flexibility, as it is subdivided in many regulative modules that preside over specific processes and functions [52], the regulation of each individual gene is constrained to those of the other genes within the same module, and the modules are, in turn, closely regulated to work in a strictly coordinated manner [50,51]. Thus, the expression of each co-expressed cohort of genes is under the control of a regulatory protocol specific to a single seed tissue at a particular developmental time [53]. The observable states of the overall regulatory network represent, therefore, a tiny portion of all the possible combinations of expression states for individual genes [41].

Although the development state of the seed is connoted by its metabolic state, the latter is largely determined by translation, which, in turn, is determined by transcription. Ultimately, therefore, it is transcription that enables the actualization of phenotype from genotype, based on external and internal clues [52]. In addition, transcription is controlled by the regulatory network. The state of the regulatory network, therefore, usually is the primary controller of phenotype and development [52]. Thus, the state of the seed is ideally represented by the state of its overall regulatory network [52]. This is why, henceforth, the latter will be considered as the embodiment of the seed’s development state.

## 3. The Role of ABA in Seed Dormancy

Plant hormones are signalling molecules, physiologically active in the nanomolar to micromolar concentration range, and involved in all phases of the plant life cycle. They are characterized by being able to move across cells, thereby transferring regulatory signals far away from where they have been synthesized [54]. Intriguingly, plant hormone signalling is based on the de-repression of specific functions, and which functions a given hormone de-represses is not univocal, but it depends on the specific tissue and stage of development [54,55,56].

As plant hormones can move among cells, they act as coordinators of every cellular process within and between tissues, governing the cell regulatory networks [54,57] to make the latter synchronized and harmonized both within a tissue (wherein they must be relatively uniform) and among tissues (so that they work collaboratively). Thereby, plant hormones control all aspects of plant growth, development, adaptation to the environment, and interactions with other organisms [54]. They also regulate the dormant and germinative states of seeds [5]. Hence, plant hormones hierarchically coordinate the overall regulatory network of the seed. The opposite is also necessarily true; that is, the metabolic homeostasis of plant hormones must be controlled by transcription factors in response to various environmental signals [56]. Thus, there need to be three different functions to orchestrate an efficient response [46,56,58]: an input (sensor) function through which external clues are translated in terms of informational entries readable by the internal regulative system; an integrating function that integrates external clues with the internal state and determines a suitable response by regulating ABA and GAs biosynthesis and sensitivity; and an operative response function that orchestrates the overall metabolism according to the settlement of plant hormones. The first function is provided by receptors of environmental cues; the second and third functions are provided by the regulatory network, though they are conceptually, and operationally, different functions.

Although no plant hormone acts in isolation [54], abscisic acid (ABA) is known to have particularly important functions in seed dormancy [5,59]. Indeed, many mutations that affect ABA biosynthesis, sensing, and signalling show reduced seed dormancy and early germination [60]. Nevertheless, its exact role in this development state is still under research since ABA is not quantitatively related to the depth of dormancy [19,59,61] and there is no evidence of a preeminent role of ABA in seed dormancy at the transcriptional level [55]. In addition, the endogenous ABA content of both non-dormant and dormant seeds rapidly declines upon imbibition during the early phase of germination [61,62]. The fact that ABA has many roles in regulating growth, development, and the response to environmental stresses [60] greatly complicates this matter. Figure 2 shows a working model for the role of ABA, originally proposed to describe modulation of physiological dormancy in red rice [61]; but it applies to arabidopsis and other species as well, as it is based on conserved physiological functions.

As previously mentioned, absolute ABA level is not representative of the germinative/dormant state of a seed [19,61,63]. Seed dormancy, indeed, is sharply related to seed ABA sensitivity but not to ABA content [9,13,61,64,65]. To this regard, Trewavas [66] pointed out that plant hormone contents often do not correlate with the physiological state they are supposed to control, and that hormone sensitivity is typically much more explicative. Dormancy relieving by afterripening is, indeed, associated with differential regulation of phospho-signalling pathways leading to a decay of ABA signalling once the seed is imbibed [67]. It is, thus, competence to respond accordingly to the seed state that dictates the physiological response [57].

ABA is synthesized inside cells and accumulates in the symplast due to the ion-trap mechanism (effect 1 in Figure 2) [61,68]. In imbibed seeds, its biosynthesis can occur in the cotyledons (in dicots), the living endosperm (the aleurone in monocots), and within the embryo axis [5]. At physiological pH levels, strong accumulation of ABA inside the cells makes the diffusive movement through the apoplast negligible beyond a short distance [68]. In addition, transport via plasmodesmata plays an insignificant role in the long-distance transport of ABA—and GAs—inside a seed [68]. It not even clear whether ABA can move through plasmodesmata within the symplast of dormant seeds, since, in bud dormancy, ABA blocks intercellular communication through plasmodesmata [69]; although this might be a bud-specific behaviour. However, even if ABA could move through plasmodesmata within the symplast, this would be possible only within each tissue (or, better, within symplastic domains), as communication between some tissues via plasmodesmata subsides during seed development [70]. Thus, ABA mostly moves through the apoplast, and only close to the site of biosynthesis (effect 2 in Figure 2) [68,71]. Hence, ABA has to be exported to the outside of cells after biosynthesis, a process for which a key role of transporters has increasingly been acknowledged [72]. Notably, plant hormones do not have an easy access to the dead starchy endosperm of mature endospermic seeds, which, otherwise, would act as an enormous sink, subtracting most plant hormones from living tissues [68].

The primary action of ABA in controlling germination consists of reducing the extensibility of the cell wall, thereby increasing the minimum threshold of hydrostatic pressure necessary for the embryo cells to expand [73,74]. Correspondingly, the role of ABA in seed dormancy is to prevent loosening of embryo cell walls and of the seed covering structures that surround the embryo [7], thereby inhibiting the expansive growth of cells (effect 3 in Figure 2) associated with radicle emergence (effect 4 in Figure 2).

Interestingly, the molecular mechanism supporting ABA inhibition of growth in germinating seeds has been proposed to occur through its inhibitory action on plasmalemma proton pump H^+^-ATPase activity, which pumps protons from the cytosol into the apoplast and thus activates expansin activity resulting in cell wall loosening and cell expansion [75,76]. In this way, ABA: (i) curbs the H^+^ gradient across the plasmalemma and, thus, membrane polarization and sugar transport [28], thereby (ii) preventing the activation of cell wall modification enzymes responsible for cellular expansion [77]. In accordance with this, rice transcripts for cell wall-modifying enzymes were more expressed in non-dormant vs. dormant seeds together with some proton-transporting ATPases [55].

Cell wall acidification is a well-known mechanism in auxin-induced elongation growth [77], and it occurs in imbibing embryo axes [78,79], although it does not have a preeminent role in germination [80]. In this respect, it can be worthy to note that acidification of the incubation medium (and, thus, assumedly, of the cell walls), by itself, does not cause germination [61]. Cell elongation and events associated with reserve mobilization are, however, late events in the transition from germination to seedling growth [5].

A functioning ABA biosynthesis pathway is necessary for dormancy maintenance after seed imbibition [59]. Thus, fluridone, a carotenoid- and ABA-biosynthesis inhibitor, is effective in breaking dormancy in some species [61]. However, fluridone alone (that is, with no addition of exogenous GAs) is much less effective in eliciting germination in other species, indicating that a decline in ABA level alone is not sufficient to break dormancy, and other changes (such as synthesis of GAs and/or other plant hormones) may also be necessary [59]. In general, germination is determined by a decrease in endogenous ABA in the imbibing seed, which results from both the suppression of de novo synthesis and the activation of catabolism [58].

During seed maturation, ABA induces dormancy (effect 5 in Figure 2) [59,81], which is based on specific, but yet unknown, dormancy factors [61]. Their existence as factors distinct from ABA is inferred because: (i) in a dormant seed, the ABA level is often kept above a threshold higher than in non-dormant seeds (effect 9 in Figure 2) [61] to avert embryo expansion growth and, therefore, germination; (ii) sensitivity to ABA is higher in dormant seeds (effect 10 in Figure 2) [9,67,82], and such high sensitivity is maintained even though application of fluridone reduces the level of ABA in the seed [61]; (iii) exogenous ABA does not prevent the breaking of the seed testa, which is an early mark of germination [5,62].

ABA, on the one hand, cannot prevent germination even in dormant seeds whose dormancy has been overcome by the application of fluridone and that, thus, maintain high ABA responsiveness [61]. In dormant seeds, on the other hand, testa rupture does not occur precisely because their development is arrested (effects 7 + 8 + 23 in Figure 2). In other words, inability of exogenous ABA to fully restore dormancy in the presence of fluridone suggests that some other specific factors determine dormancy, and that, like ABA, one of these factors, but not the other, is also depleted by fluridone, since testa rupture, but not ABA synthesis/sensitivity, is promoted by fluridone and cannot be prevented by exogenous ABA [61]. As fluridone application causes a normal rupture of the seed coats [61], the development arrest caused by the dormancy factor sensitive to fluridone must also be responsible for inhibiting the hydrolytic processes that lead to the fissuring of the seed coats (effect 23 in Figure 2).

The existence of specific dormancy factors, diverse from the cytoplasmic level of ABA, is consistent with the observation that neither maternal nor exogenously applied ABA is able to induce dormancy in ABA-deficient arabidopsis seeds [63,81,83,84]. Correspondingly, proteomic and transcriptomic profiles of dormant arabidopsis seeds differ from those of non-dormant seeds treated with exogenous ABA to curb their germination and growth, indicating that the mechanism by which ABA blocks growth is different from the physiological state of developmental arrest caused by dormancy [85,86].

It is worth noticing that the dormancy factor sensitive to fluridone displayed in Figure 2 does not need to be a biologically active molecule different from ABA, it must just be an effect diverse from the overall seed (or even embryo) ABA content. In this respect, ABA compartmentalization does matter. It was shown that ABA, indeed, localizes in the cytoplasm in germinating sunflower seeds, but it localizes mainly in the nucleus in dormant seeds that do not germinate at a temperature that is not permissive for them, though it is permissive for non-dormant seeds [82]. Thus, should ABA accumulation in the nucleus be a general feature of dormancy, the dormancy factor sensitive to fluridone could turn out to be a dormancy-specific compartmentalization of ABA.

All the effects attributed to the dormancy factors, and which are not provided by applied ABA, are removed by processes that naturally break physiological dormancy (effect 11 in Figure 2). It can be worth noticing here that, apart from dormancy breaking, which occurs during afterripening of the quiescent dry seed, all the other metabolic processes that ensue from dormancy breaking happen in the seed only once it is imbibed, as the overall metabolism needs free water to be functional [28]. Hence, any action accomplished by dry afterripening is temporally separated from the realignment of the overall regulatory network and metabolism that subsequently takes place in the imbibed seed.

In this regard, noticeably, although seeds of some species display a decrease in ABA content during dry afterripening, other do not [5]. For example, dormant and non-dormant (afterripened) dry caryopses of barley have similar amounts of ABA and GAs, since dry afterripening has a weak direct effect on the amount of these hormones [87,88]. When seeds are imbibed, however, ABA decreases faster and GAs increases more in non-dormant than in dormant seeds [87]. Corresponding results were found in cultivated oat seeds [89]. In wild oats, both un-imbibed dormant and non-dormant seeds contain similar amounts of GAs, like in cultivated oat, and, following imbibition, GAs declines much faster in dormant than in non-dormant seeds [90]. In wheat, it was observed that afterripening induces changes in the seed dormancy status without altering the dynamics of ABA metabolism [91]. After imbibition, indeed, a signal specific to afterripened seeds activates ABA catabolism [21]. Even in arabidopsis ecotype Landsberg erecta (Ler), dormant and afterripened dry caryopses have similar amounts of ABA and GAs [92]. Although higher germination capacity of afterripened seeds is associated with increasing GA levels following imbibition [92], the ABA/GAs ratio is evidently more representative of the germinative capability of the seeds than individual ABA and GAs levels.

Something, therefore, changes in the dry seed that subsequently affects the levels of these hormones. Analogously, ABA content often does not significantly decrease during cold stratification of dormant seeds but only afterwards: when the stratified seeds are placed under conditions suitable for germination, they display a rapid decline in ABA content and an increase in GAs, and they germinate, which does not occur in the non-chilled seeds [5].

As seen, though ABA inhibits expansion growth (effect 3 in Figure 2), it can delay, but not prevent, a basic embryo expansion, involved in testa rupture (effect 4 in Figure 2) [61]. This implies that the development arrest that blocks germination (effect 8 in Figure 2) has been removed too. The inability of ABA to block testa rupture [5,62] is not due to a poor efficacy of effect 3 of Figure 2; rather, it is owing to effects 3 and 13 being additive [62]. In fact, once the dormancy factor sensitive to fluridone is removed, by fluridone or dry afterripening, the development arrest and, thus, the restraining effect of the seed coats are relieved too (effects 7 and 23 + 22 + 13 in Figure 2), and ABA cannot reverse this.

An antagonistic effect of embryo expansion growth on the development arrest (effect 12 in Figure 2) appears obvious when considering that, in dormant seeds, a failure of the seed covering tissues leads, more or less slowly, to promoting germination (effect 4 in Figure 2) and, necessarily, to breaking the development arrest that blocks germination [93,94]. A noticeable deferral of seedling growth even if failure (tearing) of the tissues covering the embryo has occurred [94] indicates that repression of growth expansion (effect 3 in Figure 2) is still enabled. No immediate follow-up of seedling growth also entails the lack of commitment to growth; that is, a development arrest is in force (effect 8 in Figure 2). Such an arrest must include a block to DNA replication [94], whereas DNA replication competence is required for successful germination [78,95,96].

An accidental failure of the covering tissues, therefore, does not mean that germination is physiologically accomplished: the latter can take place only when the development arrest is overcome [94]. This is a condition different from the inhibition of loosening the testa at a predetermined site (effects 23 + 22 + 13 in Figure 2), since the constraint to expansion provided by the seed coats has been relieved once coats have failed. Eventually, if, notwithstanding ABA inhibition of the expansive growth of cells, expansion happens (as the failure of the structures covering the embryo causes a drop in the mechanical restraint that opposes to embryo thrust), ABA capability of inducing dormancy (effect 6 in Figure 2) is suppressed, and the arrest of development (effect 12 in Figure 2) is overridden. When, indeed, a developmental trajectory is accidentally undertaken, the regulatory network has to follow, as they must match. This can also happen because of mutations that reduce the mechanical resistance of the covering structures [97].

In general, the seed covering structures, typically the seed coats, exert a key resistance against embryo expansion (effect 13 in Figure 2) [7], and, therefore, they have an essential role in maintaining seed dormancy. As a result, in some cases, physiological dormancy has been interpreted as “coat-imposed dormancy” (or testa- and endosperm-imposed dormancy, in dicot seeds). Although seed dormancy can appear as a combination of embryonic dormancy and coat-imposed dormancy, the latter owing to multiple cell layers surrounding the embryo and preventing radicle protrusion [32], all seed covering structures, even in non-dormant seeds and seeds of non-dormant species, exert a mechanical restraint to embryo expansion, often the same as in dormant seeds, but the latter either germinate or not depending on whether physiological dormancy is absent or present [6].

Indeed, the balance between dormancy and germination mechanically results from the equilibrium between physical restrictions imposed by the embryo-surrounding tissues (testa and endosperm) and the ability of the embryo to grow and protrude [98]. An increase in the embryo thrust [7] and the decline in the mechanical resistance of the tissues that cover the embryo are, therefore, the physiological events that lead to germination [98]. Hence, mechanical dormancy (i.e., physical restriction to radicle emergence by embryo covering layers) has been argued to be a mere component of physiological dormancy, with the latter being the regulated process that keeps the embryo at a low growth potential and blocks the proactive breaking of the covering tissues [11]. A specific effect of the seed covering structures exists, instead, in seeds with physical dormancy, which have a water impermeable seed (or fruit) coat, a phenomenon that does not occur in species with physiological dormancy [11].

Thus, though any tissue that exerts a mechanical restraint to the expansion of the embryo is expected to enforce dormancy [5,97], the covering layers have a pre-established, passive role, acting as an external mechanical constraint to embryo expansion [7]. The proactive loosening of these seed structures covering the embryo, however, contributes to the germinative potential [7,62]. It is, therefore, the physiological overcoming of the development arrest that is responsible for the commitment to germination and, afterwards, the natural splitting of the seed tissues covering the embryo. This is typically accompanied by a stronger embryo thrust that leads to a positive germinative potential [7]: changes in cell wall extensibility, produced via active cell wall loosening or stiffening, are the basic mediators controlling cell growth. Long ago it was highlighted that the intensity of the embryo thrust is the key factor that characterizes early seed germination vs. dormancy [99].

Biochemical dissolution of covering tissues above the embryo, however, is what allows for early radicle/shoot outward appearance, rather than the pushing action of the protruding radicle that tears them [62]. In any case, proactive regulation of physiological dormancy is chiefly based on the seed physiological response, though previous effects altering the resistance of the covering tissues can have a role too, as seed coat thickness and the deposition of seed coat polymers such as tannin and suberin affect the depth of seed dormancy and are regulated through the interaction of the genotype and the environment [21].

Along its natural developmental trajectory, germination (completed with the visible protrusion of the radicle from the seed coats) proceeds into seedling growth (effect 14 in Figure 2). Cell expansion, at this stage, is accompanied by cell division [5], whereas, in the dormant state, the cell cycle is suppressed [21]. Cell size, particularly in meristems, is coupled with DNA content, so that progression through the G1/S phase of cell cycle is adjusted to cell size according to the cell type, physiological state, and species [100]. Thus, as meristematic cells expand, they duplicate their DNA and, thereafter, they start to divide only when their DNA has been fully duplicated and their mature size is appropriate [100]. In non-meristematic tissues, plant cell expansion is driven by increased vacuolar size [101] over a stable nuclear-to-cytoplasmic ratio [102]. In most species (and in normal conditions), indeed, no cell division is observed before germination, though genome duplication and activation of cell cycle genes occur at late germination stages, contributing to germination speed [98].

For several hours prior to visible germination, the seed can re-dry without losing viability. Seed priming, indeed, consists of re-drying and briefly storing seeds to speed up their germination and improve the uniformity of emergence when they are subsequently imbibed for settling a crop [22]. As drying tolerance is a feature of the maturing seed, it means that a seed can be re-dried until its metabolism has decisively moved to a germinative trajectory in accordance with the shift in the overall regulatory network from a maturation to a germination phase.

Even seedling growth can still be arrested by ABA (effect 15 in Figure 2) in conditions of water deficit (or sub-optimal temperatures), particularly within a short development window [103]. However, except for a few hydropedetic species that survive drying even after the coleorhiza and/or coleoptile has emerged [6], if a seedling is fully dried, it dies because the specialized tissues and metabolism of the seed, the plant stage deputed to survive drying, have irreversibly changed. So, the ABA-induced arrest of seedling growth falls back in the category of stress responses characteristic of ABA acting as a stress hormone [60,78,103,104]. Blockage of early seedling growth is specifically caused by extracellular ABA [105], and, therefore, it is a different effect with respect to effect 3 (Figure 2), even though it still acts by repressing growth expansion [78,103].

Next, the peculiar role of the ABA–GAs crosstalk is examined.

## 4. ABA/GAs Antagonism and the Hormone-Balance Theory

The antagonism between ABA and GAs has been long known to be a typical aspect of the regulation of seed development, with ABA prevailing in dormant seeds, whereas GAs dominate in germinating ones [5,30,58]. Correspondingly, ABA reduces and GAs promote the seed germinative potential [7]. Their respective actions in regulating dormancy and germination can, thus, be modelled in terms of slowing or speeding up the time to germination as their concentrations exceed thresholds that vary both deterministically, in accordance with the original average dormancy status of a given seed lot and in response to conditions such as chilling or afterripening, as well as randomly among seeds of the same lot [22,74,106,107].

Although, in nature, ABA has no molecular variants, more than 130 natural GAs have been identified [108]. GA_1_, GA_3_, GA_4_, and GA_7_ are the main bioactive gibberellins, and differences in biosynthesis and sensitivity are large among species, tissues, and developmental stages [108]. Thus, ‘GAs’ is used here to indicate gibberellins as a group of bioactive molecules.

GAs promote germination by triggering responses that enhance the growth potential of the embryo, both through relieving the restraints imposed by ABA as well as by overcoming the mechanical constraints provided by the covering layers surrounding the embryo, thereby lowering the growth potential threshold required for germination [58,97]. The latter is, however, a rather late event in germination; that is, it happens at a stage very close to radicle emergence [104]. Thus, GAs, even though required for the completion of germination, are not directly involved in many processes taking place during early germination, such as the initial mobilization of seed storage proteins and lipids [104].

Noticeably, spatial separation of responses to ABA and GAs within a seed embryo shows that crosstalk between ABA and GA is non-cell-autonomous and is controlled at the level of hormone movement between the spatially separated signalling centres [108,109]. In this respect, it is also worth noticing that hormone response, and not local hormone abundance, defines the site of these signalling centres [57,109]. This highlights the importance of considering the state of the seed overall regulatory network (meant as the ensemble of the regulatory networks of all the homogeneous cell groups) as the proper regulatory level to connote the seed state.

Analogously to ABA—and auxin—GAs are acids, and they too are, therefore, subject to the ion-trap mechanism [108]. This makes exogenously applied GAs quickly effective. However, GAs display lower accumulation in the symplast with respect to ABA, which favours apoplastic diffusion of GAs over ABA [68]. Thus, even though the ion-trap mechanism limits GAs ability to move out of cells, GA movements between symplastic domains of the seed are physiologically modulable by regulating GA efflux transporters more easily than occurs for ABA [68].

If dormancy is broken and ABA levels fall, then GAs production in the embryo is de-repressed to stimulate water uptake through vacuolisation, as well as cell wall loosening and endoreduplication in specialized tissues [21]. This is obviously linked to their respective roles: growth arrest for ABA and growth for GAs [5,56]. They effectively represent the champion hormones associated with one or the other of the two possible development trajectories of a seed. It is, indeed, the balance between ABA and GA signalling that underpins the seed germination potential, rather than one or the other hormone alone [40].

The ABA–GAs balance, accordingly, is a central regulatory feature that integrates multiple interactions among environmental cues [5,110], and it controls cycling through dormant states [12,39]. Even if their interaction is quite complex and also ethylene and brassinosteroids are antagonistic to ABA [5,56], the relationship between ABA and GAs can be conveniently resumed as a reciprocal repression (effect 16 in Figure 2) [58].

Although ethylene production is not an absolute requirement for dormancy breaking and seed germination, its effects greatly vary among species [111]. Ethylene, indeed, is a germination stimulant rather than a physiological regulator of seed dormancy [21,112]. It counterbalances ABA inhibitory effects [111], whereas ABA limits ethylene action by down-regulating its biosynthesis [5].

Auxin and jasmonate, on the other hand, stimulate ABA functions and enhance dormancy [113,114], at least when the seed is competent to respond according to a physiological state of dormancy. Specifically, auxin induces hypersensitivity of seeds to ABA and thereby inhibits germination in dormant seeds, whereas afterripening induces transcriptional repression of specific auxin signalling genes [91]. Exogenous auxin represses seed germination also through increasing the ABA/GAs ratio [115].

Intriguingly, auxin is involved in maintaining embryonic identity in the developing seed [116]. Thus, higher expression of genes for auxin biosynthesis in red rice dormant seeds imbibed for 8 h with respect to non-dormant ones [55] might be associated with the active maintenance of embryonic identity. This latter is transitory during early imbibition in non-dormant seeds, whereas a resting embryo persists indefinitely in dormant seeds [39,104].

It is worth noticing that high ABA levels inhibit auxin biosynthesis and auxin-related regulatory pathways in seedlings [117]: opposite effects of ABA on auxin in dormant seeds and in seedlings could well be due to a diverse response—that is, a diverse competence—in the different developmental phases. When seedling growth starts, indeed, the maturation program has already ended, and embryonic identity has been overthrown. In general, dormant and non-dormant seeds display the up-regulation of specific subsets of auxin-responsive genes, indicating that they differ in auxin transcriptional regulation [55]. Specific cross-talking with other plant hormones evidently determines the actual response in each physiological condition [55] according to developmental competence.

ABA and GAs do not directly interact to antagonize each other, but, rather, they act through the expression of genes involved in the synthesis, degradation and response of both hormones [109]. In general, there is mutual down-regulation between these two hormones: ABA down-regulates GAs metabolism and signal transduction, while GAs reciprocally subdue ABA metabolism and signal transduction [5]. Specifically, the model of Topham et al. [109] includes stimulative effects of ABA on both its own synthesis and degradation as well as a repressive effect on GAs synthesis, whereas the response to GAs would stimulate ABA degradation while inhibiting both ABA and their own synthesis. As the net effect of this complex crosstalk is ultimately defined as antagonistic, and its full dynamics are not yet established, I display it as a reciprocal repression (effect 16 in Figure 2).

In addition, as the ABA to GAs ratio is the most obvious single value to represent the balance between these two plant hormones in the context of seed dormancy, I will use it throughout this review (its reciprocal would make sense as well for germination, but, as the focus here is on ABA, the use of the ABA/GAs ratio seems obvious).

Although the crosstalk between ABA and GAs is complex, some aspects are fundamental. Firstly, as seen, plant hormones signalling is based on the repression of repressors of transcriptional activators, and the specific effects of the plant hormones depend upon the active transcription factors and repressors involved in each given response [55,56]. This mechanism is also stabilized via proteolysis of the repressed factor: the ubiquitin-mediated degradation of regulatory proteins is involved in all of the plant hormone response pathways [54,55]. Specifically, ABA negatively regulates Protein Phosphatases 2C (PP2Cs), which in the absence of ABA repress Sucrose-Non-Fermenting-1-Related protein Kinases 2 (SnRK2s) activity. Thereby, ABA promotes SnRK2s phosphorylation of ABA-responsive element-binding factor proteins (ABFs) and, thus, activates the ABA signalling pathway triggering the response to ABA [56].

Analogously, GAs elicit degradation of DELLA repressors (which act as negative regulators of GA signalling), thereby activating the GA response promoting cell expansion and proliferation [56]. In addition, when GAs levels are low, DELLAs stimulate the expression of genes that enhance ABA accumulation and signalling [58]. Thus, germination requires GA-induced degradation of DELLA proteins, which are important regulators of the ABA/GAs crosstalk [58]. Later in germination, further degradation of DELLAs by GAs also activates the transcription of α-amylase in the aleurone of barley and rice seeds [58].

As remarked above, the ABA/GAs antagonism comprises two main conceptual layers of regulation [56]: in a first layer, which above I called the integrating function, the ABA/GAs antagonism determines and tunes up the metabolic homeostasis of ABA and GAs and is controlled by the regulatory network in accordance with the development stage and environmental and seasonal signals; at a second layer, which I called the operative response function, GAs and ABA antagonistically control growth according to cues of development and stress, through interactions between ABA and GAs signalling components that mediate, and finely attune, the ABA/GAs antagonistic relationship. The former function implies that two separate routes of the metabolic homeostasis of ABA and GAs lead to opposite patterns of ABA and GAs accumulation, with antagonist effects. At the latter layer, instead, ABA/GAs crosstalk orchestrates a rapid and efficient response to developmental changes by modulating growth according to current environmental conditions. This leads to an effectively operating hormone homeostasis balance for regulating plant growth [56].

The architecture of the input (sensor) function that conveys external clues into the regulatory network is complex and variable across species, and, in arabidopsis, it includes DOG1, which intervenes at both main layers of ABA/GAs antagonism regulation [58]. In this respect, it should be noted that Hilhorst and Karssen [118] concluded that the endogenous action of ABA is comparable to exogenous germination-inhibiting conditions like osmotic stress, so that induction of dormancy by ABA in imbibed seeds might just be the result of its inhibiting germination. However, in the control of dormancy, ABA acts a developmental regulator too, whereas, under stress conditions, ABA just operates as a stress hormone. Different states of the overall regulatory network must therefore be associated with these two different roles. And a diverse competence of the seed state must dictate which role ABA exerts and when.

The classical hormone-balance theory assumes that compounds that either inhibit (e.g., ABA) or stimulate (e.g., GAs) germination are simultaneously present, and, depending on whether the former or the latter prevail in the balance, dormancy or germination is promoted [30]. Karssen and Laçka [119] proposed a revision of this hypothesis for seed dormancy, stressing that ABA and GAs are expected to act at different times and sites, where ‘expectance’ implicitly hints at developmental ‘competence’. For example, ABA induces dormancy during maturation, and GAs play a key role in dormancy release and in the promotion of germination. Thus, a shift in the ABA/GAs balance enables the transition from the maturation to germination trajectory [120]. Although, in fact, ABA has a role in maintaining dormancy in the imbibed seed (Figure 2), this is owing to the resting phase being extended from seed maturation to the imbibed seed, partially superseding the switch in the developmental programme caused by desiccation [8,59,104].

The revised hormone-balance theory, in any case, should not be seen as GAs and ABA being fully exclusive, since some amount of each hormone is typically present (at least as close precursors of bioactive forms) even if the other is predominant [56,87,89,92,120]. Indeed, an ABA/GAs crosstalk can take place only if they act at the same time [58]. The hormone-balance theory, hence, can be conveniently represented in terms of the ABA/GAs ratio, which defines the seed developmental state [109].

Very interestingly, mutual inhibition (effect 16 in Figure 2) is a general motif of regulatory network architecture that controls binary branch points between two mutually exclusive development states [41]. Such regulatory circuit is called bistable [41,48] and is wholly coherent with the known opposing relationship of ABA and GAs [40,109,121].

## 5. The Binary Development of a Seed as an Example of Bistability

Bistability is the condition of a system that has only two relatively stable states, whereas all other theoretically possible states are not stable and, therefore, quickly turn into either one or the other stable state. This condition appears to be at the heart of decisive biological phenomena [48]. The binary development of a seed is an example of bistability [40,109,122]. A double-negative feedback loop (Figure 3A), also called a mutually inhibitory network, can generate bistability [48]. A bistable circuit consists of two regulative elements, and each can become predominant if its initial level is higher (Figure 3B) or its repression (by the other regulative element) is lower (Figure 3C) than the other. This corresponds very well to the interplay of ABA and GAs in the seeds as envisioned by the hormone-balance theory.

This basic circuit is overall instable if no element predominates (Figure 3D), as only negative reciprocal effects exist. This setback is overcome if one or both regulative elements have a positive feedback loop, which produces a tristable circuit (Figure 3E). Counter-intuitively, in fact, adding positive feedback loops into the architecture of a bistable circuit changes the dynamics of the regulatory network so that the intermediate state becomes able to display a promiscuous gene expression into a locally metastable state [41]. The stronger the self-enhancing effects the more stable the intermediate state is [123].

Unfortunately, positive feedback loops also cause the predominant element to grow up indefinitely if the system moves to one of the two stable states rather than settling into the metastable state (Figure 3F,G), even though the existence of these loops avoids problems of overall instability at equilibrium (Figure 3H). Thus, although in a biological system, some additional dampener must be present in this kind of circuits to curb the level of the predominating element to a physiological limit (as discussed later), the presence of self-enhancing effects can be useful to generate intermediate metastable states.

If, according to the hormone-balance theory, we see ABA and GAs as the eminent representatives of alternative, competing states of the overall regulatory network, their mutual inhibition (effect 16 in Figure 2) makes them match the description of a bistable [40,109,121], or tristable, circuit (Figure 3). The ABA–GAs balance in seeds is indeed characterized by a rapid amplification of the prevailing effect: GAs trigger an increase in GA content and inhibit ABA production and signal transduction, whereas ABA stimulates its synthesis and signal transduction by causing a decline in GAs [5].

Although positive feedback loops of ABA on its own biosynthetic pathway have been described in arabidopsis seedlings [124], genes related to ABA metabolism are up-regulated by exogenous ABA in seedlings but not much in seeds [125]. In the latter, therefore, positive feedback loops can be mediated by the (still hypothetical) dormancy factors and the mutual inhibition with GAs (Figure 2).

It is important to notice that a regulatory circuit needs the initial levels of the regulative elements as input. This raises the problem that, during seed imbibition, hormonal control relying on genetic protocols can be executed only when physiological and organizational conditions have re-established [126]. What happens during the short decisional window in the first hours of imbibition [39] when the seed is in an unstable transitional state [40] is, therefore, particularly interesting. Specifically, the first 8 h of imbibition represent a ‘decision phase’ during which the developmental program, either germination or dormancy, is settled on [39]. Selective translation of mRNAs specific to either process will then occur depending on the outcome of this early ‘resolution’ [39]. Correspondingly, dormant and non-dormant seeds cannot be transcriptionally distinguished in the dry state, but only shortly after the initiation of imbibition, that is, after the ‘decision phase’ has been resolved [39,43,127]. Abley et al. [121] refer to this phase as a non-germinating state—which presumably corresponds to the maturation resting embryo program that maintains embryonic identity during early imbibition [104]—after which non-dormant seeds transition to the germination steady state concomitantly with a rise in GAs production.

During the passage from this stage to the (re-)activation of a functional regulatory network, stochastic fluctuations would result in variable germination capabilities and times [121]. However, stochastic fluctuations cannot explain changes in the dormancy level deterministically caused by dry afterripening. Instead, seeds respond rapidly after rehydration to the changes that occurred in the dry seed during afterripening, indicating that afterripening pre-sets the transcriptional response that follows the initiation of imbibition [127].

Therefore, although the overall regulatory network governs all the seed physiological responses and, thus, germination competence is determined by the current transcriptomic state of the imbibed seed [43], the initial input cannot be directly provided by the circuit itself (which coordinates the overall regulatory network), since this circularity would leave the starting state of the circuit (i.e., when the seed is imbibed) either in a state of indeterminacy, or stuck in a permanently fixed state. The former outcome is quite obvious: should the regulative circuit start from an equilibrium condition by default (owing to the reset caused by desiccation), the fate of the system would, then, entirely depend on random fluctuations, with no deterministic fixing. On the other hand, germination would never occur if the ABA/GAs balance were itself the determinant of the seed’s fate because GAs are never predominant in the dry seed, since the maturation of the drying seed on the plant is driven by ABA, whose preponderance is then fixed as the metabolism halts.

If, instead, some other internal input decides the initial levels of the regulative elements, the problem of circularity is solved by temporally separating the ‘decisional’ and operative mechanisms: (i) the dormancy factors are chiefly modulated during seed development, moist chilling, and dry afterripening, whereas the ABA/GAs circuit governs germination/dormancy in the imbibed seed; that is, the self-enhancing feedback (effects 5 + 9 + 10 in Figure 2) is split in two segments working at different phases (that is, they are temporally separated); (ii) even if the dormancy factors may be modulated in the imbibed seed at the same time as the ABA/GAs circuit operates (since germination and induction of dormancy can be concurrent processes), the timeframe of such modulation is slower than that over which the circuit operates on the current regulation of germination [4], so that the latter adapts to an input that is slowly changing but is not a direct echo of itself.

This, of course, implies that the current regulation of germination and the setting of dormancy are managed by different modules of the overall regulatory network. This fits well with the separation of the integrating and operative response functions described above and corresponding to the two layers of regulation evidenced by Liu and Hou [56]. In the case of dormancy regulation, as said, it is additionally expected that these two layers of regulation operate with different timeframes. Hence, though the bistable circuit characterized by the ABA/GAs balance does not decide by itself the ultimate developmental trajectory of a seed, it unequivocally characterizes the overall regulatory network, and it is for this reason that it has been widely acknowledged as a basic indicator of the fate of an imbibed seed.

## 6. ABA/GAs Antagonism in the Context of Bistability

Despite the well-known role of GAs in stimulating the use of the main seed reserves for growth (effect 18 in Figure 2) being a post-germinative event, supporting seedling growth (effect 19 in Figure 2) rather than germination sensu stricto [5], GAs are also involved in triggering the loosening of the cell wall and, thus, expansion growth (effect 17 in Figure 2), as well as the fissuring of the seed coats (effect 22 in Figure 2) and, therefore, radicle emergence (effect 4 in Figure 2). This dual action of endogenous GAs on germination was highlighted by Karssen et al. [128].

Regarding the promotive action on embryo growth, the activation of GA-responsive genes induces cell wall-remodelling enzymes, which play a critical role in germination by enabling embryo cell expansion [62,129,130,131,132]. Active GAs typically increase after seed imbibition and cause the loosening of cell walls to allow cell expansion and division, as well as weakening of the covering layers above the embryo to enable radicle protrusion [133,134,135]. However, accumulation of GAs is not causally related to dormancy breaking; rather, it is closely associated with germination [87]. In accordance with this, Karssen et al. [128] concluded that, in the imbibed seed, the synthesis and sensitivity of GAs, which determine the growth potential of the embryo, are controlled by the degree of dormancy, which is initially set up by ABA during seed development. The physical events linked to embryo growth are, indeed, the effects of an overall regulatory network committed to germination, wherein, therefore, the binary fate of the seed has already been decided (for example, by afterripening the dry seed), and the bistable circuit, which operates in the imbibed seed, needs to have already been actualized according to the established trajectory. How the latter becomes established is the ultimate question.

ABA inhibits the mobilization of the main seed reserves through its repressive effect on GAs (effect 16 in Figure 2) [5]. As dormant seeds are metabolically active [5], though not growing, they need to consume some reserves to maintain the basal metabolism. This might be possible because ABA does not inhibit storage lipid mobilization in the endosperm, although lipids stored in the endosperm are typically used to fuel seedling establishment [93].

Apart from ABA antagonism of GA-induced mobilization of the main seed reserves, glucose itself has nuanced effects on the balance between the two plant hormones. In non-dormant seeds, on the one hand, exogenous glucose (at concentrations that do not support an osmotic effect) delays germination by suppressing ABA catabolism [136,137]. On the other hand, exogenous ABA delays germination and inhibits the GA-promoted mobilization of the main seed reserves, causing an inhibition of seedling growth, but this inhibition can be alleviated with sugar treatment [28,125,138,139]. Although glucose can also relieve ABA inhibition of expansin genes during late germination [125], exogenous glucose cannot relieve seed dormancy [61]. The latter finding is in agreement with the observation that a low reducing sugar level in embryos of dormant grains (associated with the presence of starch granules) is not a cause (and, thus, it probably is an effect) of dormancy [140].

It is possible that all these effects are linked to the fact that GAs are also required for germination but are repressed by glucose [125]: it seems probable that such repression is strong during early imbibition (prior to the maturation program that maintains embryonic identity is overstepped), so that applied glucose promotes ABA dominance; whereas it ought to be weak during late germination and seedling growth (when GAs are physiologically activate to mobilize soluble sugars), so that exogenous sugars would rather overcome ABA inhibition and favour GAs prevalence (even though they can block further synthesis of GAs, in a negative feedback regulation). Thus, the sugar effects could be opposite depending on the timing of the treatment because of a diverse response over different development phases. Again, developmental competence is determinant.

In dormant seeds (wherein the maturation program that maintains embryonic identity protracts indefinitely), metabolism displays interesting similarities with the power-saving metabolic protocol induced in dormant buds, which closely resembles the ‘low energy syndrome’ (typical of stress conditions) aimed at saving carbon use to support essential maintenance functions rather than additional growth, which is, therefore, arrested [141]. This metabolic program involves down-regulation of sucrose-induced and ribosome-encoding genes, as well as of genes related to cell division and anabolism, while basal fluxes of carbon skeletons and energy are obtained from sources other than sucrose, like amino acids, lipids, and proteins [141]. ABA is deeply involved with this syndrome, as it causes down-regulation of cell cycle genes and induces expression of the *INHIBITOR OF CDK*, which arrests the cell cycle in the G1/S phase, in which dormant plant cells are typically found. ABA also antagonizes PP2Cas, the phosphatases that negatively regulate SnRK1, which coordinates energy balance, metabolism, and growth. Thereby, ABA boosts SnRK1 activity, which is, instead, repressed by sugars and trehalose 6-phosphate [141].

Quite interestingly, in some cases, the arrested developmental state of a dormant seed can be overcome by exogenously providing a high concentration of GAs [6,142]. This could be due to such a large amount of GAs being able to subvert the bistable circuit not only because of its inhibitory effect on ABA, but also owing to the excess of GAs being able to start cell wall loosening and seed coats fissuring. The latter, analogously to the failure of the seed coats, can elicit germination by suppressing any further induction of dormancy by ABA (effect 6 in Figure 2), and more importantly, by partially overriding the development arrest (effect 12 in Figure 2), thereby pushing the seed toward germination (which, once attained, is irreversible).

The promotion of the germination percentage and rate via treatment with GAs is a widespread effect, and its efficacy and applicability increase as the depth of physiological dormancy decreases, across species, from the deep (wherein it is usually inefficacious) to nondeep level (wherein it is typically quite effective) [6]. This is a non-physiological intervention that, like the depletion of ABA by fluridone (an inhibitor of ABA synthesis), stimulates the dormant seed to germinate, sometimes even substituting the physiological breakage of dormancy carried out by moist chilling or dry afterripening [6].

Unlike stratification or afterripening, applied GAs do not stimulate the germination of fully dormant seeds unless exceedingly high concentrations are used [142]. This is owing to the above-mentioned fact that GAs are mostly active in the promotion of germination and not in the breaking of dormancy per se [118]. Artificially flushing the seeds with GAs, however, is particularly effective (because of effects 17 + 4, 17 + 12 + 8 and 6 in Figure 2) in speeding up germination in partially dormant seeds that were already fully able to germinate albeit slowly [128].

Applied GAs may, indeed, speed up mobilization of starch reserves, an effect largely exploited in the barley malting industry [5]. This well-known effect of GAs is commonly observed because these seeds are already destined to germinate, and the bistable circuit has only to be strengthened into the germinative trajectory.

GAs are usually much less effective in promoting germination when the seeds are fully dormant, even in a species with nondeep physiological dormancy [61,143], because—in the mature, imbibed seed—the bistable ABA/GAs circuit is not the main determinant of dormancy, but, rather, its state is determined by the still unknown dormancy factors (Figure 2). A commitment of these factors to full dormancy can be envisaged as a powerful resistance to germination even if the bistable ABA/GAs balance is temporarily subverted by exogenous intervention. Partial germination can ensue if the GAs flush is able to lead less dormant seeds throughout the germinative trajectory on a short timeframe, overcoming the capability of the dormancy factors to reverse this effect, so that only the more strongly dormant seeds are maintained in a development arrest. Were it not for some specific dormancy factor that rules over the ABA/GAs balance [61,144,145], exogenous addition of GAs would ineluctably lead to overriding dormancy, and deep physiological dormancy would not be even possible.

It is interesting to note that, during seed development, the bistable ABA/GAs circuit is the main determinant that causes dormancy (Figure 2, effects 16 + 5). Correspondingly, high levels of exogenous GAs prevent the onset of seed dormancy in this phase [146].

Remarkably, the reverse attempt is ineffective: high concentrations of exogenous ABA are incapable of restoring dormancy in fully non-dormant seeds, at least not in two weeks [61]. As ABA enters the cells, it blocks germination even in favourable conditions, but the seeds will germinate as soon as exogenous ABA is removed [61]. This ought to be linked to the longer timeframe of dormancy induction with respect to germination. How, in this regard, endogenous ABA can allow the induction of dormancy simply by inhibiting germination [118] is not clear, but again, the development context, i.e., competence, is different in the two instances. It seems, therefore, probable that, during seed development, the seed ABA/GAs balance acts as an effector of some hierarchically higher determinant. This, in any case, could be expected, since the ABA/GAs balance affects the overall regulatory network throughout the whole life of the plant: the effect that such balance specifies in every tissue at each given stage of the plant cycle must be dictated by stage- and tissue-specific endogenous determinants.

Flushing the seed with ABA negatively affects seedling growth both directly (effect 15 in Figure 2) and indirectly by repressing GAs (effects 16 + 18 + 19 in Figure 2). Direct (i.e., not mediated by GAs) inhibitory effects of the ABA signalling cascade on GA-inducible expression of genes involved in the solubilization of the seed reserves (that is, on α-amylase) is also known, further complicating the crosstalk between ABA and GAs [58]. An overall direct effect of ABA on GA-induced starch solubilization is not shown in Figure 2 because it is not yet clear whether apoplastic or symplastic ABA is responsible for this post-germinative event.

Although Karssen et al. [128] have shown, through the use of GA-deficient mutants of arabidopsis and tomato, that GAs are absolutely required for germination, seeds of GA-deficient and ABA-deficient double mutants are capable of germinating without application of GAs, since the absence of ABA abolishes an absolute requirement of GAs [5]. Correspondingly, use of a GA-biosynthesis inhibitor showed that the high capability for GAs biosynthesis observed in imbibing non-dormant seeds is not a prerequisite for germination [90]. Hence, a promotive role of GAs is strictly needed only if ABA has inactivated the expression of germination genes and activated genes of the maturation phase during late embryogenesis [5,78,147]. So, on the one hand, GAs are needed for triggering degradation of the growth-repressing DELLA proteins to facilitate sustained growth of the seedling [58]. On the other hand, there are at least two independent pathways inhibiting seed germination–one through GAs and DELLA and the other through SMAX1—and overcoming either one may be sufficient for germination [148]. Thus, the GAs requirement for germination would, indeed, seem not to be absolute.

Although the above discussion highlights the importance of understanding the roles of ABA and GAs in terms of a bistable or tristable circuit to improve our insight of seed dormancy, to be biologically meaningful as a conceptual model, the tristable circuit needs further implementations, whose usefulness can, however, be better appraised within the framework of the development landscape. The latter will then be introduced next.

## 7. Development Landscapes for the Seed Fate

As seen, the seed ABA/GAs balance supervises the overall regulatory network, which, in turn, presides over the metabolism and its changes across development. In an imbibed seed, this chiefly means it manages the reversible switch between dormancy and germination. An intuitive way to visualize prefixed alternatives in the development of biological systems is the development landscape (Figure 4), which is a stability landscape applied to development. In this section, I will show that the development landscape is an impressive way to illustrate how the overall regulatory network, and thus the ABA/GAs balance, can be used to represent the state of the seed along its development trajectory.

A stability landscape depicts the possible changes in a complex system (most commonly either an ecosystem or a cell) that can exist in many different states with varying probabilistic stability [48,149]. A full representation of these states would require them to be described as points on an informational space with a huge number of dimensions [48]. Each theoretically possible state is, indeed, identified as a combination of many state variables, e.g., soil/water/meteorological features and frequencies of species in an ecosystem, and the set of intracellular concentrations of all the biochemical species, or, as an alternative, the expression levels of all genes, in a cell.

In the stability landscape, every state of the system is represented as a point on the *x*-axis; the closer two points are, the more similar the two corresponding states are. The *x*-axis represents, therefore, the diversity of states. However, a huge reduction in dimensionality is necessary to represent points lying in a high-dimensional informational space on a mono-dimensional axis [48]. Thus, this condensed representation is mostly used for conceptual purposes. Nevertheless, such a reduction is objectively sensible when the *x*-axis is thought to correspond to a ‘representative’ output of the regulatory system that drives development [48]. The *y*-axis represents the variable through which changes in the system occur [48]. It is commonly envisioned as the development time, though a more sophisticated time-related variable has been proposed [123]. The *z*-axis, i.e., the height of the plot, is oppositely linked to the probability of the state and is commonly represented as a fictitious form of ‘potential’ [48,123], though, more properly, it is a measure of improbability.

Similarly to physical systems, the improbability associated with each state tends to shift the system to a state of lower improbability according to the restraints and constraints that shape the macroscopic dynamics of that system, including any living organism [150]. Complex systems, indeed, are commonly subject to several constraints, external as well as internal. Their states can, therefore, have a higher or lower probability of appearance that translates inversely into the elevation of the landscape [123]. In this way, peaks and ridges represent states of lower probability, whereas valley bottoms correspond to states of higher probability. Development, indeed, can occur only through pre-stablished valleys, typically declining toward different development fates, so that a biological system “slides” toward its fate like a ball would roll down through a slope (Figure 4). Each valley in the landscape represents, therefore, a possible developmental fate (i.e., a trajectory), and the ridges between the valleys maintain the development fate once it has been undertaken [48].

Developmental trajectories are ‘canalized’ to provide consistent outcomes despite genetic and environmental perturbations [41,48]. The overall plot is, thus, a surface plot of improbability, over which the system tends to shift from a state of high improbability (and, therefore, lower stability) to one of lower improbability (i.e., more stable) according to the determinants that shape the landscape. Compellingly, living organisms can develop along a path toward lower developmental improbability (which ensures a consistent formation of the phenotype from the genotype) even though their growth and development are associated with a decreasing thermodynamic probability, thanks to their capability of exploiting existing disequilibria to make their own existence possible [150]. Differently from developmental probability, however, assessing the thermodynamic probability would require considering the thermodynamic state of the system plus its surroundings, as we are not dealing with a closed system [150].

Depending on its developmental probability (not on the thermodynamic one) and the local conformation of the landscape, each seed state can, thus, be envisaged as having a defined level of developmental stability at each given time of development. These kind of plots are used to illustrate the evolving scenarios for the studied system. They may, therefore, be referred to as a ‘development landscape’, a term that, in any case, is typically used to describe the development of a cell or an organism rather than that of an ecosystem.

For a cell, or cell system, the cell states are typically assumed to be defined by the states of its regulatory network according to the regulatory constraints defined by the cell protocols. If these states are considered through development, when they are mostly constrained by epigenetic states, the most famous form used for this representation is the Waddington’s epigenetic landscape, wherein a cell progresses from an undifferentiated state, which is deemed to be unstable (at least, once development has started), to one of several discrete, distinct, and differentiated cell fates [48]. The Waddington’s landscape has offered an intuitive framework to conceptualize changes in the gene expression pattern, which, in principle, represent the dynamics of a system of gene regulatory interactions that impose constraints to, and drive, cell development [41,48,123].

In a broad sense, the development landscape represents the phenotypic changes in a cell, or an organism, through development, where the latter is determined by the genotype, or by the interaction of the genotype with the environment (in a seed, for example). It is worthy to note, in this respect, that development is chiefly determined by the genotype in animals, whereas more plastic rather than strictly canalized development, controlled by meristems, takes place in plants, which respond to external challenges and opportunities via growth and development [57].

As said, the *x*-axis corresponds to a representative output of the regulatory system that drives development, even correlatively. This can be, for example, a phenotypic measure of the trait under study, or a weighted index of the intracellular concentrations of some key biochemical species, or a combination of relative levels of expression of a set of genes assumed to be representative of the whole regulatory network (at least in the studied phase of development), or it can be the ratio of the levels of the hormones that coordinate the overall regulatory network.

The ABA/GAs ratio is an obvious way to epitomize the seed state. A combination of expression levels of genes associated with this ratio might also be used. For example, Abley et al. [121] proposed, but did not test, a combination of *DELLAs*, *ABI4*, and *ABI5* expression levels to characterize seed bistability. Krzyszton et al. [43] used gene expression signature values of two groups of co-expressed genes showing largely opposing patterns of expression—namely, germination-associated and dry-seed-associated genes—to quantify germination competence. Footitt et al. [38] considered the relative levels of expression of two genes, *AHG1* and *DOG1*: an increase in their ratio coincided with increasing germination potential in the population. Topham et al. [109], though using the expression of specific genes as markers of the several functions involved in quantifying their ABA/GAs model, eventually employed the sensitivity of the system to GAs for modulating the level of dormancy in their system, since an increase in GAs sensitivity is associated with the progressive loss of dormancy, either during afterripening [151] or low temperature treatment [152]. In any case, it should be noted that the different scales of choice might not be equivalent: as the phenotype is not a linear function of the transcriptome or of plant hormones’ content and sensitivity, a phenotypic trait is not necessarily linearly related to the ABA/GAs ratio or to the relative expression levels of a set of genes. Nevertheless, the different scales ought to display an ordinal relationship.

The development landscape provides a straightforward visualization of the dichotomous development trajectories of living seeds upon imbibition (Figure 5A–C). Different conditions of the seed can be topographically mapped (Figure 5D–F). In this regard, it is worthy to note that seeds that, along the germinative trajectory, attain the state of commitment to germination (cg in Figure 5D), but do not enter phase III of germination, are still in a condition that allows a two-way switch (that is, the seed can switch back and forth between the germinative and dormant states). This switch marks the transition of a seed from dormant to germinable, and it is, therefore, an important developmental checkpoint [22]. In other words, this initial phase of germination is a reversible process in which the resting embryo program of the maturation phase can be resumed if an osmotic stress, or another restraint of germination, takes place [14,104,153]. Indeed, stress-related gene expression can be promoted by dormancy-inducing conditions even in the absence of abiotic stress [12]. In this respect, the first 8 h of imbibition, approximately, comprise a short window of time in which ABA-induced genes that are typically expressed during seed maturation are transcribed and start to be translated, if germination is restrained [62,103,104].

A prolongation of the resting embryo program, or of physiological mechanisms related to it, during early imbibition is also consistent with diverse transcriptional hints pointing to a central role of the proplastid in seed dormancy [12,55], since the plastid has been implicated in the regulation of the germination potential [104]. The upregulation of the photosynthetic machinery may also be a reflection of the seed’s commitment to germinate in anticipation of autotrophic growth [104]. In accordance with this, GAs control the biogenesis of chloroplasts in developing seedlings, and when the content of GAs in germinating seeds is low, DELLAs block the conversion from proplastids to chloroplasts [154].

The dormant seed is not transcriptionally static, and the transcripts present late during phase II of water uptake are no longer merely those accumulated during development on the mother plant [12]. Notwithstanding sharp transcriptional differences between dormant and non-dormant seeds in phase II of germination, major patterns of mRNA stored in the dry seed are neither evidence of the degree of dormancy nor of the germination potential during the subsequent imbibition, but, rather, they reflect the developmental context of seed maturation [153]. That is, many of the abundant dry seed transcripts simply match translation during seed maturation [62]. However, out of the transcripts that have been transcribed during seed development and stored in the mature dry seed, only a subset is translated early in germination [155,156]. This means that a specific subset of transcripts that are presumably necessary during early germination is selected for translation out of the whole set of seed-stored mRNAs that had been transcribed and translated during maturation, and this subset is independent of the dormancy state of the seed [157]. Indeed, at this stage, there is no correlation between transcriptome and translatome, and a selective and dynamic recruitment of mRNAs to polysome is translationally regulated, distinctly for dormant and non-dormant seeds, only later, during phase II of imbibition [155], when a surge of transcription of germination-related genes also takes place in non-dormant seeds. This confirms that dormant and non-dormant seeds are also distinguished by a different control of transcription [157]. Selective translation of subsets of transcripts specific to dormant and non-dormant seeds during phase II of imbibition was also observed in other studies [156,158].

Concordantly, even in non-dormant seeds, *NCED*s and many other ABA-related genes are not in a pre-silenced state before germination (which would imply the resting embryo program of the maturation phase had already been suppressed), but, rather, they are repressed progressively throughout the germination and seedling-growth process, in association with a decline in ABA content [159]. A common origin of the trajectories of dormant and non-dormant seeds (Figure 5B) reflects the matching states of the overall regulatory network in the two physiological conditions (apart from eventual oxidative phenomena, which will be discussed later).

As a consequence of the extension of the maturation resting embryo program, ABA, stress, and dormancy responses significantly overlap at the transcriptome level [12,153]. Thus, ABA was suggested to regulate the germination potential also through the maintenance of embryonic identity [104]. Seed dormancy, therefore, seems to correspond to a state wherein the developing programme of the seed has not actually commuted from embryonal (like in the maturation phase) to germinative, notwithstanding seed desiccation. However, the pattern of transcription differs between newly imbibed primary dormant seeds and seeds maintained in an imbibed state for prolonged time [12]. Thus, the overall regulatory network of the imbibing dormant seed moves along a development trajectory (Figure 5C) prior to, eventually, stabilizing (d in Figure 5F).

Transcriptional differentiation between dormant and non-dormant seeds occurs, as said, shortly after the initiation of imbibition, once the seed has ‘resolved’ its development state [39,43,127]. This means that the development states of the dormant and non-dormant seeds bifurcate during phase I of water uptake (Figure 5B), but this happens immediately before the transcriptome starts mirroring this change (as the ‘decision’ precedes the operative response). The transcriptional state of the seed, therefore, could not be the perfect way to denote the developmental state of the seed. As differential translation requires, however, different states of the overall regulatory network, the state of the translatome (and, then, of the metabolome) is an even later indicator of the seed development state. Hence, the state of the overall regulatory network, though slightly deferred with respect to the commitment of the seed to either developmental state, is still the best way to characterize the latter.

During the late phase I of water uptake, thus, the dormant seed undergoes a developmental turn and heads along the dormancy trajectory to a resting basin where it stays idle as long as necessary (Figure 5C). Germinating seeds display an opposite veer (Figure 5A), which is initially reversible. Once the seed state has moved to early post-germinative growth (pg in Figure 5E), however, the system exhibits irreversibility, that is, it cannot turn back into a non-germinated seed. Any residual of the maturation program (such as an overall regulatory network befitting a resting embryo) needs to have vanished at this point.

Following imbibition, both dormant and non-dormant seeds first display a change in specific metabolites, and, shortly after (>6 h of imbibition), a large change in transcript abundances for metabolic functions, before a substantial recovery of translational activity takes place [31,104]. Pre-existing enzymes are, hence, responsible for the early seed metabolism [104]. Dramatic changes in the transcriptome can be observed upon imbibition, after as little as 1–3 h, that is, in phase I of water uptake [160]; most of these changes are, however, due to degradation of stored mRNAs [62,157,161]. One of the first processes occurring upon imbibition is, indeed, the clearing out of many stored transcripts [162]. This is presumable linked to the fact that only a minor subset of stored transcripts is used for translation during germination [156].

Meanwhile, a specific transcriptional switch point of the germination process occurs at about 3 h of seed imbibition [160], and it extends (through the first and second germination phases) to the first 9 h of incubation in barley [135] and to 6 h in arabidopsis [31]. During imbibition there is also a progressive recovery of metabolism [5,160], and the actual state of the overall regulatory network of seeds in the state of commitment to germination depends on the recovery dynamics. It can, therefore, be expected that when such a state is transient, because seeds are quickly germinating (Figure 5D), the above-mentioned changes rapidly follow one another and overlap. Differences are, thus, observed among published studies, owing to the diversity of species, exact timing, and experimental conditions. If, on the other hand, seeds linger in the state of commitment to germination owing to the inadequacy of some environmental variable (more on this later), it can be expected that they reach an advanced expression of genes for RNA processing and the translational machinery, as well as of germination-specific genes (since, as seen, the transcriptional state of the seed is not perfectly linked to its the developmental state). The state of commitment to germination is, therefore, defined by the overall regulatory network, but contextually to the actual landscape.

Although the development trajectory is most often a sequence of steady states (through valleys), basins represent local minima in the stability landscape that best match the idea of (provisional) stable states. The deeper a basin is the more this is true. Nonetheless, a basin is, indeed, a basin of states, which means that the dormancy basin (d in Figure 5F) is a set of states through which a seed fluctuates (within boundary conditions defined by the seed protocols), both deterministically, in response to small changes in the environment (such as the diurnal cycle, quick changes in ambient temperature, or localized soil stimuli), or because of stochastic oscillations of the metabolism (as phenotypic differences among individuals of a genetically uniform population may occur even if the environment is homogeneous and stable) [163]. Although individual variation within a seed population is stochastic, it is neither noise nor error; rather, such variation is an integral component of how seeds fulfil their evolutionary function of survival [22,24]. These oscillations, however, further complicate the interpretation of experimental results and, especially, their comparison among different studies.

It may be worth noticing that the contour graphs shown in Figure 5D–F are different from the 2D graphs employed by Topham et al. [109], though they are theoretically linked: whereas the former display the diversity of states (for example, in terms of the ABA/GAs ratio) on the *x*-axis, and the *y*-axis is time-related, the latter only show the diversity of states as defined by the relative ABA and GAs abundances. As, in any case the latter graphs [109] display trajectories indicating how the system shifts among states (which occurs through time) and such trajectories ultimately converge toward two stable resting states of the system (that is, basins, according to the terminology used here), the two types of graphs are obviously mapping conceptually similar things, although the relative landscapes appear as distorted representations of each other. Eventually, therefore, both kinds of graphics illustrate the bistable fate switching in the seed development: either a high-ABA, low-GA stable state representing dormant seeds, or a high-GA, low-ABA stable state representing the germinating seeds.

Even though portraying the seed state in terms of the development landscape is enticing, it raises compelling questions, too. The first one is how the landscape changes and, therefore, what reshapes it. This shall be discussed later. A second question also promptly arises when noting that, in the case of partially dormant seeds (Figure 5B), the hydrotime distribution shown in Figure 5H does not directly match with a random distribution of states around an average trajectory. A reasonable fit is, instead, observed for either fully germinating or fully dormant seeds (Figure 5G,I). A convincing explanation will require a discussion of how the development landscape is presumably shaped, and a more in-depth featuring of the metastable states, which are presented thereafter. By now, suffice it to say that metastable states allow the seeds to follow an array of individual paths that transitorily differ from the most probable trajectory (Figure 6) and better fit a random distribution even for partially dormant seeds, at least at the beginning of their trajectories, when their germination behaviour is decided (Figure 5H). Now, it may be useful to look closer at the transcriptional switch between dormancy and germination, a major determinant of the journey of a seed across the development landscape.

## 8. Transcriptional and Translational Features of Dormant and Non-Dormant Seeds

Several studies aiming at discriminating the transcriptional fingerprints of dormant and non-dormant seeds have been published, but it is not easy to achieve an unambiguous general interpretation out of them, as differences in species, timing and experimental setup muddle the results. Notwithstanding these difficulties, some important considerations can be drawn by carefully comparing findings among the literature. First, at the end of seed development, in preparation to seed desiccation (hence, in orthodox seeds), the nuclei of mature embryo cells shrink and transition to a condensed heterochromatic state, thereby repressing gene expression [164]. This process reverses when the seed imbibes and germination commences: the nuclei regain their size and chromatin de-condenses to the euchromatic state required for gene transcription [164]. Transcription starts very few hours following imbibition both in germinating and dormant seeds [5,104,153], indicating that nuclei have already transitioned back to the euchromatic state [164]. During rice germination, there is a two-step large rearrangement of the transcriptome that is caused by mRNA degradation and synthesis, and is accompanied by later changes in metabolite levels [160]. In general, as seen, this is cleaning time for the seed transcriptome [62] in preparation for big changes.

Although de novo transcription starts within a few hours of imbibition, it is not strictly needed for completion of germination [104]. Indeed, the mRNAs stored in the seed already provide a basic support for germination [78]. Nevertheless, transcription is required to ensure a suitable germination rate and is essential for a successful transition to post-germination seedling establishment [153]. Translation, instead, is necessary to accomplish germination; therefore, the regulation of germination is chiefly under translational control [153,155,165], which means that, as previously remarked, different mRNAs, specific to either the dormancy or germination trajectory, are translated [39,155,156].

It is worth noticing that in contrast to what occurs at the end of germination in preparation for seedling growth, de novo synthesis of proteins involved in the translational machinery is not immediately required during early imbibition [153], because seed-stored components of the transcriptional machinery are sufficient to ensure early gene expression [12,104], if the seed is not too aged. All components of the transcriptional machinery are, indeed, stored in dry seeds and are quickly activated upon imbibition [62]. Intense translational activity (to integrally re-build the transcriptional machinery) would, in fact, require strong energy consumption, which is restrained under the slightly hypoxic conditions occurring in the embryo axis at early imbibition [153], also because mitochondria in dry seeds need repair and differentiation before restoring full ATP production by oxidative phosphorylation, especially in afterripened and aged seeds [62].

However, reactivation of basal mitochondrial activity occurs quickly upon imbibition, whereas mitochondrial biogenesis requires some time for recovery [126]. Correspondingly, a transient increase (between 1 and 6 h of imbibition) in abundance of transcripts for organelle biogenesis in general, and mitochondrial biogenesis in particular, is a crucial and common feature of early seed germination [31,160,162]. In dormant seeds, early transcriptional activation also aims at recovering plastid functionality [55].

When a seed is committed to germinating, it needs to be prepared for germination and growth, which will require a more intense metabolic activity than that occurring in the dormant seed [28]. Thus, a non-dormant seed must be ready to translate mRNAs required for the initiation of germination when conditions become fully permissive for germination [166]. Necessarily, translational activation precedes ribosome biogenesis, which starts during the seed to seedling transition [165]. Genes associated with RNA processing and the translational machinery are, therefore, stored themselves and up-regulated prior to the completion of germination, so that intense translation is specifically promoted in the embryo of germinating seeds [167].

Translational up-regulation implies the mRNAs associate with more ribosomes (thus forming polysomes), since the intensity of translation usually correlates with ribosome density along the mRNAs [98]. Moreover, translational up-regulation requires that the translational machinery is up-regulated first, at least in arabidopsis [166]. As seen, this happens when the seed are committed to germinating, prior to the expression of germination-specific genes. Accordingly, the up-regulation of translation-related genes has been suggested to be an early hallmark of germination in arabidopsis [43,166]. Conversely, fully dormant seeds do not show a surge of gene expression associated with protein synthesis since they are not preparing to germinate [12]. Hence, in moist soil, a greater abundance of translation-related transcripts is the most noticeable difference between arabidopsis seeds committed to germinating and dormant seeds through the dormancy cycle [12,166]. A stably moist soil is required because ribosomal protein gene expression and ribosomal activity do not occur during the initial, fast water uptake (phase I), but they increase dramatically during the plateau phase of imbibition (phase II), facilitating the de novo synthesis of proteins important for seed germination [165].

Although the up-regulation of translation-related genes is an early hallmark of germination in arabidopsis, it has been remarked that whereas mRNAs for the translational machinery are already abundant in the dry seed of both arabidopsis and barley, the expression of genes encoding ribosomal proteins further increases early during germination (<12 h) in arabidopsis, but it rather decreases during barley germination [31]. This suggests that while barley has already accumulated abundant transcripts for translation genes during the maturation phase, their accumulation is partially delayed to early germination in arabidopsis. A diverse investment in protein synthesis machinery between arabidopsis and barley during maturation indicates that different strategies are pursued in these species to achieve germination [31]. A surge in mRNAs for the translation machinery is, therefore, useful to tell apart germination and dormancy in small oil seeds but not in large starchy caryopses. It does not seem, thus, to be a universal hallmark to distinguish dormant and non-dormant seeds. Furthermore, basic physiological differences occurring between seeds of different species (for example, oil and starchy seeds, whose main reserves are stored in the cotyledons and the endosperm, respectively) are reflected in diversity of transcriptional patterns [31]. In addition, small seeds have a limited capacity to support heterotrophic growth, and thus, there needs to be a quick up-regulation of the machinery required to support autotrophic growth, including photosynthesis-associated genes [31]. Species with small seeds (like arabidopsis and most weeds), in addition, are more at risk of dying for starvation if the seedling does not emerge from the soil quickly, that is, if they are buried too deep [6]. These seeds need, thus, to block germination if lack of light indicates that the soil surface is not close [168]. Their nitrogen reserves are slim too, so that enough nitrate must be present in the soil for they to germinate safely.

During seed germination, changes in polysome occupancy, indicating changes in the set of genes that are translated, are restricted to two temporal phases [165], one encompassing seed hydration (in the first 6 h of imbibition) and one seed germination (between 1 and 2 days of incubation in water). These two translational shifts display a noteworthy difference: genes that are transcribed and those that are translated overlap to a great extent during seed hydration, but they largely differ during the seed to seedling transition due to translational control [165]. On the one hand, correspondence of genes that are transcribed with those that are translated is observed as a natural consequence of mRNA-dependent translation when development proceeds quick and smooth, since gene expression shows how a biological system is preparing to change its metabolism and regulation [55]. On the other hand, translational control is required when a developmental switch is encountered, because an overall modification of transcription requires a thorough adjustment of the proteins regulating it.

Beyond translation, Howell et al. [160] found that by considering a time lag (of a number of hours) between the transcript and metabolite changes, there was a good correlation between changes at the two levels in germinating seeds. This holds true even when considering dormant and non-dormant seeds [158]. However, this concordance is invalidated when a development shift is enacted in conditions that do not allow for an immediate realignment of the overall regulatory network, for example, when the resting embryo phase that should extend from seed maturation to the imbibed dormant seed is broken by dry afterripening to relieve dormancy. Another instance of misalignment between transcriptome and metabolome occurs, as we will see, when metastable states of a partially dormant seed lot converge into the most stable germinative trajectory during the seed to seedling transition, a change in the overall regulatory network that must be shepherded via translational control.

Several germination-specific genes, like those associated with cell wall modification, are expressed just before testa rupture [166]. Hence, they characterize phase II of the seed germinative development and have been proposed as early markers of physiological germination prior to any morphological marker such as the rupture of the seed coats [55,62]. So, a surge in the expression of these genes marks the state of commitment to germination (cg in Figure 5D), which still allows a two-way switch, rather than marking the onset of phase III of germination.

Although translational capacity has been invoked to play a major role in the switch from the dormant to non-dormant state, at least in arabidopsis [12,166], upon imbibition, as seen, overall translational activity of stored mRNAs is similar in both dormant and non-dormant seeds [153]. Hence, a greater abundance of translation-related transcripts in germinating seeds is a late event in the transition from dormancy to germination; actually, it just happens when the seeds are committed to germinating (cg in Figure 5D). These seeds, indeed, prepare to kick-start active translation following radicle protrusion [153]. How the overall regulatory network of a seed is set up to follow a given trajectory so that only the corresponding mRNAs are translated is the missing step.

A second transcriptional switch (after the master switch toward either the dormant or germinative trajectory) occurs in the late germination phase, and is associated with the developmental transition from germination to seedling growth [135,165]. As novel transcripts must be translated for this transition to become effective, a fully functional translational machinery is necessary for seedling growth [153].

Now, we can go back to the development landscape and see how it is reshaped according to environmental conditions.

## 9. Shaping the Dormancy Landscape

In a development landscape, the system state can move by dynamical driving forces, by stochastic fluctuations, or by alterations of the landscape [41,123,149]. A dynamical driving force like downhill sliding cannot explain how the system state move from one basin to the other, by definition. Moderate stochastic fluctuations of the seed overall regulatory network are an essential feature of the development landscape, as they avert a system from permanently settling down into local shallow minima and, thus, help prevent regulatory networks from being trapped in phenotypic states that can be temporarily useful but sub-optimal in the long run [163]. However, they are not the main cause of large movements between deep basins, as random fluctuations are dampened by ‘canalization’. External stimuli, through the sensor and integrating functions, can, on the other hand, change the development landscape: they can tilt the landscape to either the right or the left, so that valleys and basins can be eliminated from a landscape and subsequently recreated through a reverse of the process [48]. This means that the genotype x environment interaction reshapes the landscape according to the seed’s protocols, and, in the imbibed seed, this must take place by means of the overall regulatory network.

A problem of circularity promptly arises: as the development landscape establishes the probabilistic trajectories for major changes in the overall regulatory network, it cannot be for the latter to decide the former, on pain of either full indeterminacy (which, would entail a random germinative response of the overall seed population, not just a random distribution of germination potentials around a deterministic average, which is a real feature of seed dormancy) or immutability (because the seed state would be stuck in a causal loop) of the system, as seen above for the initial setting. The separation of the integrating and operative response functions solves this problem too.

Thus, the overall regulatory network delineates the developmental landscape based on external clues and according to internal protocols. But this needs to be performed independently of the current state of the operative response layer of the regulatory network. This means that there must be a developmental separation between the shaping of the development landscape and the regulatory network following its trajectory on it. This development separation may be in time, for example, when dormancy is induced in developing seeds, whereby pre-establishing the development landscape for when the seed will imbibe. Or it can be simultaneous but controlled by an independent regulative module, for example, when partially dormant seeds germinating under unfavourable conditions gradually shift to secondary dormancy, moving the developmental landscape from one with intermediate probability of the dormancy trajectory to one with a higher probability. In any case, the separation of the overall regulatory network in two distinct functions allows them to operate with different timeframes, even concurrently. The correspondence of this features with the outline of the revised hormone-balance theory is striking.

Separation, i.e., causal independence, of the establishment of the developmental landscape from its operativity is revealed by the effect of dry afterripening: proactive changes in the regulatory network are not possible in the low hydrated seeds, as the cytoplasm of seed cells becomes a highly viscous glass that severely limits molecular diffusion and the availability of free water, so that transcription and translation do not occur [5,153,169,170]. Nonetheless, dry afterripening changes the dormancy status of the seed [5,26]. This is key evidence that a metabolic, rather than transcriptional, effector is the true specific determinant of dormancy. Long ago, Roberts [171] remarked that this must be associated with a chemical, rather, than biochemical reaction. Specifically, non-enzymatic oxidations are possible in dry seeds, and they represent the most plausible explanation for the effect of dry afterripening on development [28].

The separation, or lag, between establishing the development landscape and its effect on the development trajectory requires, indeed, a tuneable “memory” or “setting” that establishes what the regulatory network can do (that is, it determines the development landscape), even though the regulatory network can itself act on such setting, when and how external stimuli induce it to do so (in the imbibed seed), like plausibly happens during stratification.

As some internal determinant must decide the initial levels of the regulative elements (that is, it determines the first layer of regulation to enact the ABA/GAs balance), this same effector may be used to stabilize the intermediate metastable steady state (Figure 3H). For example, by incorporating the determinant mechanism within the self-enhancing effect (so that self-enhancing is buffered by the slow kinetics of change in the determinant and tuned by external clues); which is what the dormancy factors do in Figure 2. To this aim, the determinant of this dormancy setting must, indeed, be included into the regulation of the circuit, so that the circuit is adjusted to the proper level, set by such determinant. This corresponds to the ‘reference signal’ outlined by Trewavas [46] as enmeshed with the stage of development: a reference, or setting, that the integrating function uses to assess the current development trajectory. In other words, the ABA/GAs balance would act as an homeostatic regulator of the seed physiological state within a feedback system to maintain the dormancy/germinative trajectory according to an independently determined reference value [57].

The functional model (Figure 2) shows, indeed, that the dormancy factor(s) modulate both ABA synthesis and ABA sensitivity. The dormancy factors are, in addition, tuned by ABA (effect 5 in Figure 2), but these reciprocal effects are intended to take place, as said, at different development times (namely, seed imbibition and seed development), or with a diverse timeframe. In general, the tuning of the dormancy setting should display slower dynamics of modification (i.e., persist longer) with respect to the time required for unrestrained germination, since, as remarked above, it must operate like a “memory”. For example, secondary dormancy can be induced under unfavourable germination conditions as these prolong over time. Were this setting not slow adjusting, daily fluctuations in temperature could completely change the dormancy status of the seeds, with strongly erratic effects on germination, rather than in synchrony with the cycle of seasons. Thus, the overall regulatory network manages everything in the imbibed seed, but it operates over two distinct timeframes, so that the current development occurs in a well-defined, already-established, developmental landscape, while, at the same time, the next developmental landscape is set up.

To exist as separate functions, as seen, these two pursuits of the overall regulatory network need, in addition to different regulative modules, a “memory” that must be set outside the overall regulatory network. The obvious candidate for this role is, of course, epigenetic modification. Epigenetic states are, indeed, a well-known “memory” of the regulatory network [172], since epigenetic modifications constitute a slow fixation process for the stabilization of the cell state [35]. In addition, several genes associated with chromatin remodelling have been suggested to regulate seed dormancy and/or germination, though they chiefly intervene either during seed development or in the repression of seed maturation genes during the transition from seed to seedling [159,173,174]. Specifically, genome-wide demethylation occurs in the germinating seedling at the onset of cell division [175].

Unfortunately, epigenetic modification cannot occur in the dry state, when afterripening takes place. Furthermore, according to Huang [41], the idea that chromatin modifications are the primary cause of specific gene expression patterns, operating upstream of the transcription factors by controlling their access to DNA target sites, must be reversed: since the expression pattern of the transcription factors is a consequence of the control of the regulatory network, the sites of chromatin opening and closing should mirror the dynamics of the gene expression profile. Thus, chromatin modification should not be considered the chief coordinator, but, rather, it would serve as an additional, important layer of stabilization of the expression pattern, as established by the network of transcriptional regulation [41]. This ought to apply to stratification as well, because in several instances, the ABA/GAs balance does not change during moist chilling, but, rather, it adjusts to the level of dormancy determined by stratification once the seed is placed under conditions suitable for germination [5].

Even though epigenetic modification induces slow landscape changes, it leads to the stabilization of states by deepening the valleys, through a positive feedback reinforcement between gene expression and epigenetic change [35]. Such epigenetic feedback reinforcement locks the system into a given developmental route. In mature non-dormant imbibed seeds, for example, the chief aim of chromatin remodelling and histone modification is the stable repression of the transcription of the seed-maturation genes, which is an essential feature of the switch of the seed developmental program to ensure normal seedling morphology since the protein encoded by these genes play a predominant role in triggering and maintaining an embryonic cell fate [174]. Chromatin modification, indeed, typically ensures homeorhesis of development, that is, the stability of the developmental time-course of a system state against external perturbations [35]. This is just the opposite of what dormancy cycle requires: high flexibility and reversibility—at least, until the seed germinates. This collimates with the above-mentioned propensity of plants for a more plastic and less canalized development [57]. In fact, histone genes are up-regulated in the late germination phase [135], and whereas chromatin decondenses dramatically in the first two days after imbibition, it increases in compaction during early seedling establishment and vegetative growth [164], that is, once stable canalization of development has occurred.

A robust canalization of seed development along either one or the other trajectory would also prevent the spread across both the development routes of an array of metastable seed states (Figure 7) originated from stochastic variability in the initial seed conditions and leading to a random distribution of individual paths for partially dormant seeds (Figure 5H). Some histone modifications, nonetheless, change dynamically with the depth of dormancy in the imbibed seeds of the soil seedbank, suggesting a role for epigenetic modification in smoothing environmental signals to determine changes in the depth of dormancy [176]. This might favour, in the long run, the synchronization of dormancy with the seasonal cycle. In any case, epigenetic modification does not seem to be the best candidate for the role of dormancy factor, even if it stabilizes the diversity of states of the overall regulatory network between germinating and dormant seeds as well as across degrees of dormancy depth by deepening the valley or basin the seeds move to.

Thus, a determinant of the seed’s state must steer both the network of transcriptional regulators (first) and chromatin modifications (in consequence) coherently toward the corresponding development state. Definitely, there must be some primary reference that resolves the direction of these coordinated changes [46]. Hence, both chromatin modifications and the expression pattern of the transcription factors ought to be regarded as effects, rather than causes, of the primary mechanism underpinning seed dormancy.

The subordinate role of the overall regulatory network (with respect to an unknown dormancy determinant) reiterated throughout this review would be exceptionable only if selective oxidation of stored mRNAs were demonstrated to be the cause of dormancy. Regulated changes in gene expression are absent in dry seeds [169], but selective oxidation of stored mRNAs occurs during dry afterripening concomitantly with the release of dormancy [153]. Thus, differences in transcript stability purportedly cause a relative change in the expression profile such that some mRNAs associated with germination and growth increase their relative abundance while most residual mRNAs decline in abundance [153,161]. In this way, it is supposed, they prevent subsequent translation of proteins that function in the suppression of germination after imbibition [153].

It ought to be noticed, therefore, that apparent up-regulation of genes can be observed during dry afterripening, but this is due to an overall decay in the mRNA pool that causes an increase in the proportion of those genes that are more stable [153,157]. Even if there is hardly any change in the observed abundance of transcripts between dry dormant and afterripened seeds, the oxidation of specific mRNAs during afterripening might reduce their translation during seed imbibition, even if they are still present in the transcriptome [98]. Thereby, there would be a selective recruitment of mRNAs into polysomes during imbibition, and the inhibition of germination will be either maintained or not at the time of imbibition depending on the oxidative imprinting of the seed [98]. During the following imbibition, the effects of proteins translated from mRNAs supporting dormancy are assumed to override those supporting germination in the absence of dry afterripening, whereas this dominance would be relieved by afterripening because the former mRNAs display a lower incorporation in stable monosome complexes [153].

Differences in the stability of specific mRNA have, thus, been explained in terms of mRNAs supporting germination, and not those supporting dormancy, being targeted to monosome complexes during seed development, which increases mRNA stability [153]. However, most mRNAs that are transcribed during seed maturation are stored as monosomes in the dry seed, but only about one sixth of them are translationally up-regulated during seed germination [156]. A role of reactive oxygen species (ROS) has been suggested [177], but it is not even exactly known how oxidation of stored mRNAs actually happens. Great uncertainty persists over this process.

A diverse explanation was proposed by Oracz et al. [178]: ROS could trigger protein carbonylation to release dormancy. Carbonylation, indeed, occurs during afterripening and may play an important role in the transition from the dormant to non-dormant state in dry seeds, as carbonylated proteins are degraded by the 20S proteasome [28,39]. Neither of these mechanisms, however, has been definitively confirmed. Ultimately, the mechanism by which oxidative imprinting in dry seeds is converted to molecular signals is still unknown [98].

Selectively damaged mRNAs failing to be translated and degradation of carbonylated proteins represent plausible modifications that take place in the dry seed as a consequence of the non-enzymatic generation of reactive oxygen and nitrogen species [28]. It is, however, not clear how mRNA degradation, or protein carbonylation, both caused by ROS, can be highly selective: extensive degradation of mRNAs has been observed in consequence of dry afterripening [55], and ‘extensive’ usually does not match with ‘selective’.

If, in any case, the release of seed dormancy were indeed caused by the deterioration of specific subsets of stored mRNAs during afterripening, the seed expression pattern should indeed be regarded as the primary mechanism underpinning seed dormancy, and its selective changes as the cause of the transition from dormancy to germination. This mechanism, however, would be specific to dry afterripening, so that other dormancy-breaking treatments, like stratification, would require a completely different mechanism.

Whatever the case, even if the ABA/GAs ratio were indeed controlled by the overall regulatory network (at least, by its first layer of regulation) without being supervised by another, non-transcriptional, internal determinant, it would still retain its role as marker of the seed germinative/dormant status, and the development landscape would maintain all its illustrative utility.

## 10. Matching a Binary Fate with a Dormancy Continuum

As a sample of genetically uniform, viable seeds can show any germination percentage between 0% and 100%, it is obvious that: (i) the two development trajectories (toward either germination or dormancy) can coexist within the same environmental conditions; (ii) the germination capability of the seed population is determined by the depth of seed dormancy; and (iii) which state each individual seed ultimately reaches (around the population average) depends on random differences in the conditions the seeds were exposed to during development, dispersion, and storage [21,23], as well as on stochastic variations in the seed metabolism due to natural oscillations in the overall regulatory network within its boundary constraints [41,47,121,123,163]. Indeed, even in an isogenic population, dynamic stochastic fluctuations of phenotype can ultimately result in the heterogeneity of the population [163]. Specifically, a simulation based on a model of the ABA/GAs regulatory network showed that variability in ABA sensitivity, generated by stochastic fluctuations in the dynamics of the underlying gene regulatory network, may account for the variability in germination times observed experimentally [121].

Population-based threshold modelling of seeds germinative behaviour [26,179] assumes, indeed, that a seed batch represents a normal distribution of germinative potentials that ultimately translates into a binary response, as each seed either germinates or not, based on the specific test conditions (such as, water potential of the imbibition medium, temperature, light intensity, and so on). The distribution of germinative potentials across seeds sharing the same genotype is thus assessed as a random variable characteristic of every seed lot [26,179].

Although the fate of a seed is binary, population-based threshold models based on hydro-, thermal-, and hydrothermal time [22], account for the fact that in a partially dormant seed batch—that is, a sample of seeds that in part germinate and in part can be dormant—the seeds that germinate (even the whole sample) do germinate more slowly, even much more slowly, than seeds that had their dormancy fully relieved. In other words, seed dormancy is a quantitative trait that can go from complete absence of germination to full dormancy passing through slow, either partial or full, germination.

It must, therefore, be clear that even though the development fate of a seed is binary, its physiological status is not. In this respect, it needs to be highlighted that seed dormancy can also be ‘conditional’ (aka ‘relative’) when considered with respect to the environmental conditions; that is, seeds germinate only over a narrower range of conditions than non-dormant seeds [3,6,9]. This typically means that their dormancy state is conditioned by small differences in temperature that would not affect the germination capability of fully non-dormant seeds. Partial and conditional dormancy are, thus, different facets of the same complex phenomenon.

In this respect, a question that a development landscape helps pondering is how exactly a binary seed development matches with a dormancy continuum. Even though a seed sample can be entirely composed of seeds directed along either one or the other development trajectory (Figure 5A,C), and thus subject to the full dominance of one or the other state of the bistable circuit (Figure 3), all intermediate cases are possible (e.g., Figure 5B) even if a seed either germinates or not. In the latter case, which state a given seed’s circuit ultimately achieves depends solely on the peculiar history of that individual seed.

Figure 5G–I shows how the germinative potential (that is, the opposite of the intensity of dormancy) is continuously distributed across seeds, according to the hydrotime model [22,26]. When seeds are incubated in water, seeds that have negative base water-potential thresholds (i.e., higher germinative potential) germinate, those with positive base water-potential thresholds (i.e., lower germinative potential) do not germinate and stay dormant. This provides the binary outcome of the dormancy continuum. Moreover, seeds of the fully germinating population (Figure 5G) germinate faster than the germinating seeds (those with negative base water-potential thresholds) of the partially dormant population (Figure 5H), since the former population is shifted to the left (i.e., more negative) with respect to the latter. The trajectories of fully germinating seeds, in accordance with this, follow a steeper decline along the most probable trajectory of that side of the development landscape (Figure 5A).

Although degrees of dormancy within the fully dormant population (Figure 5I) are, at present, indistinguishable, within each germinating population seeds with more negative base water-potential thresholds (i.e., higher germinative potentials) germinate faster than seeds laying to their right on the Ψ_b_(*g*) axis. This is a phenotypic quantification of the dormancy status of the seeds that can be conveniently used along the *x*-axis of the development landscape as representative of the diversity among states. In this respect, Krzyszton et al. [43] remarked that sensitivity thresholds may be reflected by the continuous transcriptomic gradient of germination competence. Thus, the relationship between these two conceptual frameworks appears to be straightforward. Nevertheless, at this point, it is not yet fully clear how the partially dormant population (Figure 5H) is divided in two parts that follow either development trajectory, which ultimately produce either a seedling or a dormant seed. That is, how multiple individual paths converge into the valley bottom of either developmental route.

It should be first remarked that the hydrotime model considers that the distribution of base water-potential thresholds characterizing a seed population affects both the outcome (i.e., whether a seed either germinates or not) and the velocity with which germination is obtained [179], whereas the development landscape focuses on the former. Nonetheless, how the germinating part of the partially dormant population attains germination can be indirectly linked to the latter aspect (i.e., germination speed). To see how this can happen, we have to go back to the tristable circuit.

## 11. More on the Metastable State

I have previously mentioned that the overall regulatory network, and thus the ABA/GAs balance, can linger in a metastable state before turning into one of the two main development trajectories toward either germination or dormancy. I also broached the possibility of having many metastable states, slightly differing among seeds, so that every seed follows an individual path that transitorily diverges from the corresponding most probable trajectory (Figure 6). Within a seed population, indeed, germinative potentials are randomly distributed, at least at the beginning of their trajectories, that is, upon imbibition. This is possible thanks to the properties of the tristable circuit.

In a tristable circuit, any combination of levels of the regulative elements can be in a metastable state if the dynamics of the two elements equal each other (Figure 6A). As the proportion of a regulative element can be used to draw the state of a circuit (Figure 6B), it can also be used to straightforwardly represent the circuit on a development landscape (Figure 6C). The ratio of one element to the other could be used in a similar way. The main difference is that when a proportion is used, the actual values can range from 0 to 1, with equality set to 0.5, whereas if a rate is used, the actual values can go from −∞ to +∞, with equality set to 1. The former is, therefore, easier to deal with, as proportions are much more compressed toward the range extremes. Thus, the proportion is used in the figures, though the ABA/GAs ratio is referred to in the main text for the sake of simplicity.

A tristable circuit in a metastable state moves along a slope of the development landscape well above the valley bottom (Figure 6C). As the metastable state of such circuit is subject to a downhill (probabilistic) push, it is eventually destabilized (Figure 6D,E), and, thus, it ultimately swerves from its path halfway up the hill, flowing into the valley bottom (Figure 6F). Indeed, intermediate states are seen only transiently, as they are inherently unstable [48]. Correspondingly, as seen, any residual of the transcriptional program of the maturation phase, as well as of seed dormancy, must vanish when the seed starts germinating. Thus, the path shown in Figure 6F ought to be a better representation of an individual germinative trajectory followed by a seed having partial dormancy (that is, germinating slowly because of a moderate degree of dormancy) than the generic germination route indicated for these seeds in Figure 5B, which keeps to the main, most probable, trajectory.

In addition, even though the stable phenotype is robust to stochastic noise within a given environment, and it also ensures a consistent phenotypic response across environments, an intermediate phenotype is more sensitive to environmental variation [49]. It therefore has much greater flexibility in its responses across environments, thus providing the adaptability required to cope with varying environment challenges, which plants need more than animals [57]. Metastable states are, therefore, the most obvious responsible for conditional dormancy.

So, seeds that are neither fully dormant nor fully germinating (Figure 5B,H) are in metastable states that do not follow the most probable trajectory. Rather, they follow individual paths outside the main trajectory. Hence, it is the dormancy “setting” of each individual seed, and the fluctuations thereof, that act upon the overall regulatory network when it lingers, albeit temporarily, in a metastable state along a developmental landscape slope.

Eventually, the probabilistic pressure of the development landscape prevails, first gradually and then suddenly, making the overall regulatory network to “slide” toward a more stable state. Meanwhile, the initial transition state has characterized the germinative behaviour of each seed in terms of hydrotime (Figure 5G–I). As slow-germinating seeds (those with negative base water-potential thresholds in Figure 5H) ultimately become normally growing seedlings, it means that their metastable states eventually join the mainstream (most probable) trajectory at the valley bottom. Since the latter is the quickest (steeper) path of development, it follows that seeds in a metastable state (which, by remaining at a specific value of the *x*-axis, initially proceed along an almost horizontal path on the hillside, since the valley is perpendicular to the *x*-axis) have a slower motion (as the dynamical driving force is minimal). That is, their germination is slower (if they are on the germinative path: those on the dormancy path are not directly observable), in accordance with the hydrotime model.

In other words, the relationship between the hydrotime distribution and the distribution of paths on the development landscape is not linear, particularly for the partially dormant seeds. Therefore, the hydrotime distribution shown in Figure 5H does not directly match with a random distribution of states around an average trajectory in the case of partially dormant seeds because the former describes the dormancy status of the seeds, which, on the development landscape, results in a spread of the array of individual paths as the seeds pass from paths along the slope of the ridge to the two main trajectories at the bottom of the valleys.

It is worth remembering that seeds’ transcriptional heterogeneity is at least partially explained by a gradient of transcriptional competence to germinate [43]. In addition, transcriptional heterogeneity shrinks while seeds are induced into dormancy, but it widens when a partially (secondary) dormant seed population is incubated in conditions permissive for germination [43]. This means that the paths of the seeds spread across an array of metastable states. Such phenomenon could be due a narrower valley for dormancy vs. germination, corresponding to a stricter canalization.

It is, however, important to stress again that the transcriptional activity taking place before 6 h of imbibition does not determine whether seeds are able to germinate or not [153,157]. Correspondingly, by itself, the absolute value of transcriptomic germination ‘competence’ (which is not exactly the same thing as developmental competence, since, as seen, they do no overlap perfectly) does not determine whether a seed will germinate or stay dormant: non-dormant seeds imbibed for 1 h have low transcriptomic germination competence though germinate when transferred to permissive conditions [43]. This is because during early imbibition their transcriptomes still correspond to the maturation resting embryo program that maintains embryonic identity [104]. Only when this stage has been trespassed does the current transcriptomic state of the seed really come to describe germination competence, meant as the proximity of the seed population to germination. Then, transcriptomic germination competence may directly or indirectly reflect the relative position of individual seeds along with sensitivity thresholds distribution in the population as proposed in population-based thresholds models [43].

The fact that, at very early imbibition, seeds always display low transcriptomic germination competence, and a clear transcriptional cut-off between germinating and dormant seeds becomes evident only after seeds whose sensitivity thresholds are permissive for germination actually start their germination process [43], supports that a non-transcriptional dormancy effector is the determinant of the current transcriptomic state of the seed that, then, defines germination competence.

In the way it has been outlined up to this point, the tristable circuit is still non-realistic, though, because, when it deviates from a metastable state, the dominating element grows to ever higher levels owing to its self-inducing effect. To curb this effect, a mechanism must be present that fixes a maximum level for the dominating element. In biological systems, this is frequently provided by a saturation effect; that is, the self-promoting effect is constrained to be proportional to the difference between a maximum attainable level and the current level [49,180]. As expected, even a maximum threshold for intracellular ABA concentration seems to exist [105]. Thus, a self-limiting tristable circuit provides a more realistic representation of what may happen when a biological system with continuum variability faces a binary response (like in Figure 7). In this way, the level of the dormancy factors could also provide a reference for limiting the increment of the level of the predominating element (ABA, in the case of dormancy), thereby improving the stability of the steady state of a dormant seed.

A basal, minimum level (m) of an element could also be modelled by specifying it in the repression effect (for example, r_X_·[Y]·([X]-m_X_) for element X). According to the revised hormone-balance hypothesis [119], ABA and GAs ought to act at different times and sites, which means they tend to be mutually exclusive. Typically, indeed, GAs levels in ripe seeds are generally very low with respect to developing seeds [5]. Nevertheless, a minimal physiological level of each hormone is usually present even when the other is dominant, but more studies are needed to clarify this aspect.

It can also be worth noticing that although basic regulatory circuits are often modelled according to the Hill equation, which provides a flexible modelling function [48,123,180], here I wished to gradually introduce the properties of a basic circuit by using incremental steps of complexity. The Hill function, which is not well suited to this purpose, can anyway be used to model the circuit of interest within the conceptual framework illustrated here. Moreover, I omitted a term to account for the degradation of the regulative elements of the circuit, which concurs to reduce their concentrations [48]. This, and any additional term that is deemed useful, can be added to improve the reliability of the circuit with which a biological function is modelled, once the basic approach has been outlined.

## 12. Secondary Dormancy and the Final Block to Germination

Although Figure 5 displays three possible scenarios for seed batches with diverse initial levels of dormancy, an effect of dormancy induction (that is, secondary dormancy) still needs proper consideration under the development landscape framework. Environmental conditions unfavourable for germination (e.g., temperatures outside the range suitable for germination, but not promotive of dormancy removal through stratification; or a soil water potential too low for germination, but not low enough to cause dry afterripening; or the seeds are deeply buried in the soil and, thus, light and/or oxygen levels are not adequate for germination) favour the induction of deeper dormancy in seeds, typically in batches that are already partially dormant. Indeed, secondary dormancy induction is faster and deeper the greater the dormancy already is, so that it becomes undetectable in fully dormant seeds [15,181].

Unfavourable germination conditions can be visualized as a barrier to germination progress (Figure 8A). If the conditions remain unfavourable for a prolonged time, the development landscape can further change by a gradual rising of the germinative valley (now a basin; Figure 8B), which causes more and more seeds to pass from a pre-germinative state, i.e., commitment to germination [104], to a dormant one (Figure 8C), that is, secondary dormancy is induced. ABA itself, provided at a concentration that prevents germination for enough time (and in seeds that are not fully non-dormant), can allow the induction of secondary dormancy, which then persists even if the applied ABA is removed [182]. This process is reversible, since, over the seasonal cycle, a change to conditions that promote dormancy removal (typically during the unfavourable season for growth; Figure 1) causes a reversal of the development landscape to a landscape contour favouring germination.

As the environmental conditions change back and forth from unsuitable to suitable for dormancy induction through the seasons, the flip-flop of the landscape prompts a turnabout of the flow of seeds between basins (Figure 8B) and causes the seeds to gradually overflow from one basin to the other along the annual dormancy cycle of the soil seedbank (Figure 1). When conditions that break dormancy are followed by conditions favourable for germination, the seeds draw out of the germinative basin and germinate.

The barrier to germination progress (Figure 8A) can also be a convenient way to think about the contrasting views on whether dormancy relief and stimulation of germination are separate processes [4,6,19,183]. Specifically, the depth of dormancy regulates sensitivity to environmental clues (principally light and nitrate), and seeds only progress to germination completion on exposure to stimulative environment cues once they have become sensitive to such stimuli [3,38]. That is, light and nitrate trigger germination completion once dormancy has been relieved [4], and the development block that persist in their absence can be represented as a barrier to the germination progress (Figure 8A).

The promotion of germination by nitrate in light-sensitive weeds is, usually, effective only in conditions of relative, i.e., conditional, dormancy [3,6]. This means that nitrate (at ordinary concentrations) does not promote germination of fully dormant seeds, so that two distinct blocks to germination must be present in photoblastic species, like arabidopsis, wherein DOG1 would regulate what has been called a final ‘layer’ of dormancy [176]. The existence of two separate blocks to germination—with one block insensitive to nitrate, while the other is sensitive to nitrate and mediated by GAs—was also reported for wild oat [184].

It has, thus, been proposed that in arabidopsis (and other weeds), there are two dormancy mechanisms (or ‘layers’) with a temporal separation, such that deep dormancy (promoted by ABA signalling, which includes repression of GA signalling) occurs in winter, whereas shallow dormancy (due to ABA-independent DELLA repression of GA signalling) takes place in spring and summer [19]. A seed germinates only when both ‘layers’ of dormancy are removed.

In general, however, repression by DELLA proteins is absolutely required for seed dormancy and growth inhibition, and plants carrying non-functional mutations in these genes are non-dormant [21]. Nevertheless, response to light and nitrate have been shown to be chiefly related to the promotive effects of GAs on germination, rather than on dormancy per se [118]. Thus, it seems that ABA-independent DELLA repression of GA signalling is just a way for the individual seed to provide an additional control gauge, beside the dormancy-dependent ABA level, to ensure that the specific environmental conditions around the seed are suitable for germination.

On the one hand, indeed, seeds, particularly small ones, benefit from accurately sensing the suitability of their spatial position to germination [39,176]. Germination, therefore, responds to both temporal and spatial prompts [13,39]. On the other hand, dormancy setting is established in function of environmental clues (e.g., moisture and temperature) averaged across time, as it responds over a longer timeframe [4]. Although, in temperate environments, temperature is the dominant seasonal signal controlling seed dormancy of imbibed seeds [4,21], it is a poor cue to differentiate soil conditions at the single-seed scale.

Thus, two formal viewpoints exist: either dormancy cycles through a continuum of intensity, and the block to germination completion is a separate process [4,6,12] gauging germination at the single seed level, or there are two diverse mechanisms of physiological dormancy, and the seeds cycle through these two distinct blocks that are removed by different environmental clues at different times [19].

Whatever the name used for it, being a direct gauge to control germination according to localized environment conditions (that is, according to exogenous clues rather than endogenous ones), the second block to germination completion can be represented as a barrier to the germination progress on the development landscape (Figure 8A), both for light-sensitive weeds, like arabidopsis, as well as for rice and other species for which contrasting dormancy mechanisms have not been reported, although temperatures unsuitable for germination have a similar effect. Incidentally, positive photoblasticism has been observed in rice too [171,185], confirming that such phenomenon is widespread across taxa.

In any case, because of its prompt responsiveness to environmental conditions, I assume the final block to germination [19] is, indeed, a block to germination [4]. In this way, the seeds display species-specific requirements (e.g., temperature, water potential, light, and nitrate) for germination [4,6] that are dependent on their seed dormancy [3,6].

As the diverse variables can interact with each other, the response turns out to be quite complex. For example, light can compensate for a temperature unsuitable for germination in partially dormant seeds, if such a temperature is suitable for non-dormant seeds [128,186]. That is, light can alleviate a dormancy-imposed constraint onto germination, which can be seen as a barrier to germination progress (Figure 8A) independently of whether light is considered a dormancy release factor or a requirement for germination. It is probable that every environmental clue can influence the shape of the development landscape both in terms of a barrier, and by a tilt of the landscape, but with different timeframes for the two effects and a diverse efficacy for each clue.

In positively photoblastic species, once dormancy has been removed, light and nitrate further suppress ABA content and promote GA synthesis and signalling [5]. The regulation of ABA metabolism genes by light is mediated through phytochrome-interacting factors [5,40]. In general, the fact that, in light-responsive species, light alters the seed ABA/GAs balance [5,40] would support light and nitrate as being dormancy-breaking factors if one thinks that dormancy is determined by the ABA/GAs balance, but it rather supports that light and nitrate overcome a germination control if the ABA/GAs balance is, instead, envisioned as being determined by dormancy (in the imbibed seed), as shown here. In both cases, the effects of different environmental factors that break dormancy and promote germination, including dry afterripening, moist chilling, nitrate, and light, on the different gene sets are additive [5] because the ABA/GAs balance, which results from the integrating function ultimately defines the transcriptional response to the environmental and dormancy inputs.

Even the fact that, in the deeply dormant Cape Verde Islands accession of arabidopsis, nitrate can substitute for the long period (7–12 months) of dry storage or several days of cold stratification required to break dormancy is consistent with the idea that nitrate effect is targeted at the ABA/GAs balance [5]. In fact, the application of relatively high concentrations of nitrate (>7 mM) appears to have an effect analogous to flushing the seeds with exceedingly high concentrations of GAs, subverting the tristable state of the ABA/GAs circuit if enough time of incubation in nitrate solution is given for the ABA/GAs circuit to override the dormancy setting.

It is then interesting to enquire what physiological mechanism underpins secondary dormancy induction. Figure 2 shows a feedback mechanism that can account for the reinforcement of dormancy that occurs in response to unfavourable germination conditions [5,26]: the reciprocal effects 5 and 9 + 10 + 3 + 6 generate an endogenous positive feedback [61] that makes germination increasingly sensitive to external water potential and other conditions unsuitable for germination, as dormancy becomes more intense under inductive conditions [5,26].

Noticeably, GAs provide a sort of antagonistic loop (effects 17 + 6 in Figure 2). These opposing loops are expected to determine the dormancy setting when enough time is given: if the environment is stable, the two timeframes over which the overall regulatory network acts align with each other to the same commitment and the development landscape flips together with the ABA/GAs balance.

Nonetheless, as regards dormancy cycling, whereas the ABA loop is reversible, since a dormant seed can always have its dormancy relieved and germinate, the loop provided by GAs can become irreversible. In fact, the effects of GAs (namely, cell expansion and extensive hydrolysis of seed reserves) are germination and post-germination events (respectively), which, once achieved, are not reversible to dormancy, even if a flip of the ABA/GAs balance can still cause growth arrest in the seedling [103]. An earlier loop might be provided by brassinosteroids working alongside GAs [187], but the complexity of the interactions defies our current capability to understand how this concretely affects the functioning of the overall regulatory network.

An assumption implicit in the above reasoning is that secondary dormancy can be induced if the reversible ABA loop (specifically, effect 5 in Figure 2) is active. Apart from mutations that suppress such a mechanism and prevent dormancy at all, a strict instantiation of the revised hormone-balance theory would imply that when GAs become fully dominant, the ABA level could be zeroed. Should this happen, the ABA loop would be annihilated, and dormancy induction prevented. Correspondingly, Baskin and Baskin [11] remarked that it is questionable if a seed that is completely non-dormant at maturity can be induced afterward into dormancy. It seems, thus, reasonable to suppose that dormancy cycling never reaches a state of full GAs dominance. Even dry afterripening displays a decay kinetics for dormancy that approaches zero only as a limit [26]. This, of course, requires that the circuit architecture includes a minimum level for both its regulative elements, as previously discussed.

In sum, both synthesis and degradation of ABA and GAs are key metabolic features inherent to the bistable, and tristable, circuit that must be tuned according to the dormancy status of the seed for this to be reversible in the annual dormancy cycle. They are, therefore, effects, rather than causes, of the dormancy status of the seed.

## 13. ROS, NO and the Emergency Gear

Reactive oxygen species (ROS) include hydrogen peroxide and free radicals, such as singlet oxygen, superoxide, and hydroxyl radical [177,188]. At levels that are too high, ROS cause cell damage, but at moderate levels, they stimulate germination [177]. It must be noted that the different ROS have different effects, also depending on their compartmentation [177]. During germination, ROS roles vary from interaction with cytoplasmic signalling pathways during early imbibition, to cell wall weakening at the late germination phase [177]. Specifically, ROS that interact with wall polysaccharides and promote cell elongation in germinating seeds are produced in the cell walls of growing embryos at the time of radicle protrusion. ROS also have an important role in defence against attacks by challenging microorganisms [188,189].

Active regulation of ROS occurs in seeds, as ABA represses, and GAs induce, ROS production [177]. It has also been hypothesized that ROS can determine the ABA/GAs balance and, then, the dormancy/germinative state of the seed, for example, in the dry seed [177]. However, as an increase in ROS content in the imbibed seed promotes germination and is associated with ABA degradation and reduced responsiveness to ABA, with a direct effect on the hormonal ABA/GAs balance in the favour of GAs to induce germination [177], one wonders what causes an increase in ROS content in non-dormant seeds and not in the dormant ones. Indeed, at shedding, the status of dry mature orthodox seeds is largely oxidized [177]. So, how can ROS and ABA/dormancy be generically antagonists in the mature seeds if the desiccation phase is associated with both ROS generation and maintenance of primary dormancy by ABA? The effects of ROS must be highly specific and dependent on developmental competence. In the dry seeds, analogously, a purported specificity of ROS action on some mRNAs, and not others, must be explained. In this respect, protein carbonylation seems more appealing as a mechanism for dormancy release [177].

Nitric oxide (NO·) is a gaseous plant hormone involved in many plant activities [190]. It is an uncharged and lipophilic free radical that can readily diffuse across biological membranes [191]. Like for other hormones, NO· effects depend on its actual concentration [190,192]. In particular, NO· is a potent dormancy-releasing agent in many species [191,192], as well as a key mediator of the defence response to pathogen attacks in plants [190].

Dormancy breaking by NO· is mostly effective in crosstalk with ROS [191]. Interestingly, NO·-mediated dormancy release in oat caryopses is effective in both dry and pre-imbibed seeds [193], and it is associated with decreased ABA content and sensitivity, and, thus, with a lower ABA/GAs ratio [193]. In oat, NO· does not affect the content of bioactive gibberellins, and some level of ROS is required for caryopses to respond to NO· [193]. Interestingly, under conditions in which NO· and ROS are produced, NO· limits ABA signalling via tyrosine nitration, which requires both NO· and superoxide anion [194].

Dormancy alleviation via NO· is associated with a reduction in the ABA content [191], which renders the initial concentration of GAs sufficient for germination of dormant caryopses [193]. This is consistent with the previously mentioned hypothesis that increased GAs content is strictly needed only if the ABA level has not already dropped. These observations hint to a further level of ABA/GAs crosstalk by which, notwithstanding the ABA-GAs antagonism, a forced reduction in ABA does not lead to an increase in GAs. This implies that there is a regulative mechanism that detects whether a reduction in ABA is indeed ‘forced’, that is, that such a reduction does not match the seed regulatory environment that would be otherwise set to dormancy. This, again, supports that the seeds must have a dormancy setting independent of ABA, but it must be sensitive to NO·.

The postulated further level of ABA/GAs crosstalk is not shown in Figure 2 because it is not exactly known, but it may be speculated that the state of development arrest displayed in Figure 2 has additional roles, for example, inhibiting novel synthesis of bioactive GAs. If so, then a forced reduction in ABA does not lead to an increase in GAs until expansion growth overrides the development arrest both through direct and indirect (i.e., mediated by dormancy factors) effects.

Noticeably, both ROS and NO· burst after wounding of living plant tissues [190,195,196]. Seed wounding can be caused by insects grazing or can result from agricultural practices [196]. As a consequence, ROS and NO· crosstalk stimulates germination [191]. For this to occur in dormant seeds, ROS and NO· must first revoke the development arrest (effect 20 in Figure 2) and prevent ABA inhibition of growth (effect 21 in Figure 2). This mechanism is plausibly required upon wounding as an emergency response, so that the dormant seed starts growing into a seedling before its stored reserves are consumed by saprophytic microorganisms. Often, mechanical damage to the living tissues below the seed coat is sufficient to terminate dormancy and overcome the germination requirement for GAs in arabidopsis [93] and many other species [6]. Correspondingly, seed scarification is an effective method of breaking dormancy in many species [6].

In this respect, the role of ROS and NO· in breaking dormancy is an entirely different matter with respect to physiological stimulation of germination in non-dormant seeds: an ROS burst [196] is involved in the response to wounding, as the latter facilitates microorganism attack, to trigger germination before the seed rots. Even in this case, wherein ROS and NO· signalling represents a sort of emergency gear overriding the dormancy setting, the specific ROS signal intervening in dormancy breaking is not yet fully clear. The scheme of Figure 2 is, therefore, vague about this aspect. In any case, this mechanism should operate as an ‘executive’ control signal that overrides all others during a physiological emergency [57,66].

The existence of an emergency gear superseding the dormancy setting could have an important role in dormancy. If it were present in some species but not in others, it might explain the difference between deep and nondeep physiological dormancy: seeds with nondeep physiological dormancy just wake up (germinate) if someone excises the embryo from the seed, whereas seeds with deep physiological dormancy do not. A strong response to wounding in the former but not in the latter would seem to be a simple rationale for this difference.

Analogously, the fact that, in dormant arabidopsis seeds, removing both the external testa layer and the aleurone layer, but not the former alone, triggers germination, could be due not to the latter providing most germination repressive activity [197], but to the fact that the latter and not the former is a live tissue that can elicit a wounding response.

## 14. Conclusions

A high complexity of the overall regulatory network is evidently necessary for the seed metabolism to suitably respond to the environment.

Exogenous stimuli and endogenous determinants resolve the specific state of every regulatory module presiding some cellular function and, thus, the state of the overall regulatory network. The plant always responds according to protocols and through trajectories evolutionarily designed in each species to fit its environment and to proceed along a well-defined development cycle. For a seed, the master switch of such deterministic process is the toggling between dormancy and germination. Over the seasonal cycle, this translates into the dormancy cycle. All these effects can be aptly depicted by means of the development landscape, which is outlined by the genotype x environment interaction. The speed with which the shape of the development landscape changes following environmental inputs varies according to the type of response: conditions unsuitable for germination are immediately reflected as a development barrier, whereas changes in the dormancy status are slower and occur over a much longer timeframe.

The binary development state of a seed can be better comprehended if it is seen as a bistable, or tristable, circuit of key antagonistic regulative elements linked to the overall regulatory network (eminently, the plant hormones ABA and GAs), and its changes can be represented as trajectories of the ratio of the levels of these regulative elements travelling on a development landscape. The balance between ABA and GAs is the epitome of the development state in the imbibed seed (that is, it manages the metabolism according to whether the seed is going to germinate or not), with ABA dominance characterizing the dormancy state. The ABA/GAs ratio (or the proportion of either plant hormone) is, therefore, well suited for being used in these representations.

Some endogenous determinant(s) of the dormancy level must establish the ABA/GAs balance, particularly in the dry seed and, at least, in the early phase of imbibition. The model described in this review illustrates that such dormancy factors determine the germinative potential, and, therefore, whether the seed germinates or remains dormant, at least, in normal conditions. This conceptual framework is coherent and quite comprehensive, and it offers a defined and structured explanation of many features of seed dormancy that are fragmentarily described in the literature, and it is also suitable for making testable predictions.

## Figures and Tables

**Figure 1 plants-12-03963-f001:**
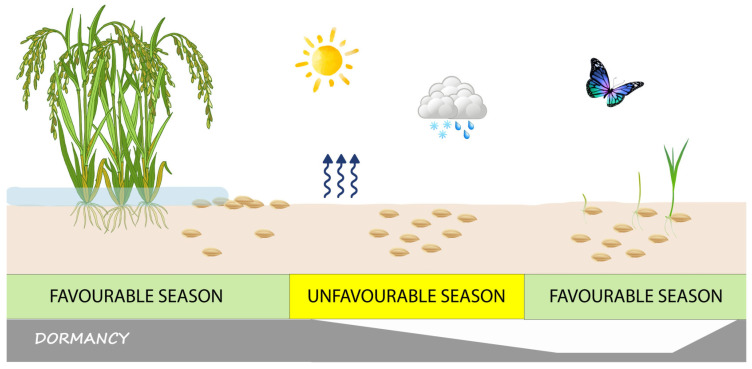
Ecological features of seed physiological dormancy over the seasonal cycle (shown in the middle pane). Seeds are shed from the mother plant onto the soil at the end of the growing season (favourable season). Various effects (e.g., tillage, trampling, waves of the eventual water cover, mulching, burrowing) lead to burying seeds in the soil. If seeds remain viable and dormant, they, together with those surviving from previous years, form the so-called soil seedbank. Sunny periods cause the soil and the seeds to dry up (waving arrows), causing afterripening. Cold months and rainfalls favour moist chilling. Thus, dormancy (lowest pane) is gradually released during the season unfavourable for growth (e.g., during dry months and/or the cold winter). Thereby, seeds come out of dormancy as the unfavourable season ends and become ready to germinate at the time of the year when conditions are favourable for the development cycle to resume (symbolized as a butterfly), usually the spring or autumn, depending on the species. Viable seeds that do not promptly germinate can re-enter dormancy.

**Figure 2 plants-12-03963-f002:**
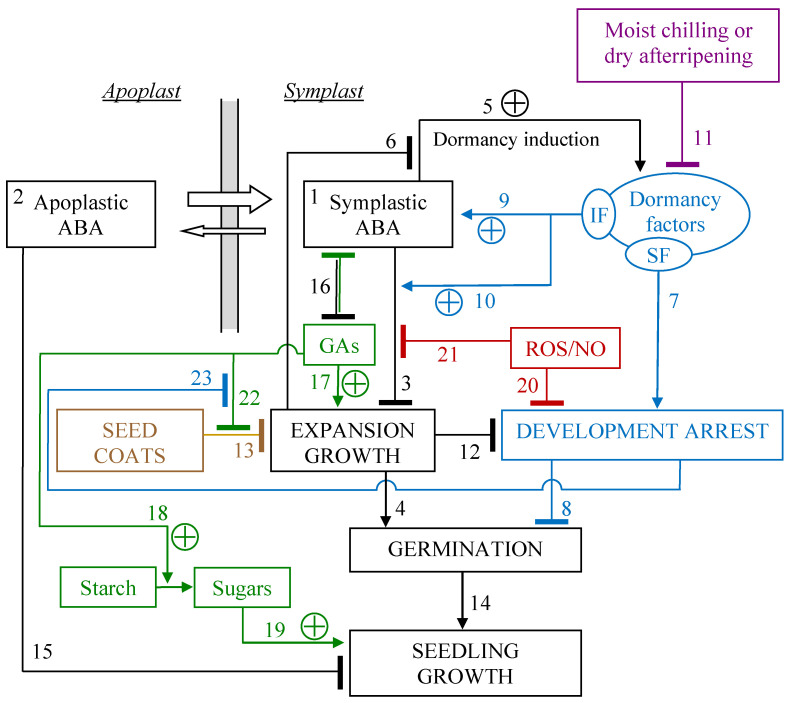
Scheme of the proposed working model for the role of ABA in seed dormancy. Six functional modules are shown: basic ABA relationships (black), dormancy factors (navy blue), dormancy breaking processes (purple), GAs crosstalk (green), seed/caryopsis coats and other covering structures (golden brown), and the emergency gear (Venetian red). Large white arrows represent ABA crossing the cell membrane, arrow lines represent causation (a cross in a circle specifies stimulation), and stopped lines indicate inhibition. (1) Basal levels of ABA are produced in the embryo axis, and (2) extracellular ABA is in a pH-dependent equilibrium with cell ABA. (3) ABA modulates cell expansion by increasing the sensitivity to external water potential, and thus, (4) it acts as a repressor of germination events associated with growth. (5) Under conditions favouring quiescence (low osmotic potential during seed ripening, sub-optimal temperatures, or lack of light for positively photoblastic species, etc.), ABA stimulates the induction of dormancy, i.e., accumulation of dormancy-specific factors, a process that (6) is suppressed by active growth. (7) A dormancy factor sensitive to fluridone action, SF, causes a development arrest, which, in turn, (8) blocks the developmental commitment to germination and also (23) prevents the breakdown of the covering structures (22) caused by hydrolytic enzymes activated by GAs in a coordinated process that promotes cell expansion and leads to testa rupture (germination). A factor insensitive to fluridone action, IF, stimulates (9) ABA synthesis and (10) responsiveness to ABA. The reciprocal effects 5 and 9 + 10 + 3 + 6 generate an endogenous positive feedback that makes germination increasingly sensitive to external water potential as dormancy becomes more intense under inductive conditions. (11) Moist chilling and/or dry afterripening remove all the dormancy-specific factors, breaking both the development arrest and the endogenous feedback that makes germination hypersensitive to water potential. Depletion of ABA and of the SF factor by fluridone disrupts the endogenous ABA feedback, and de-represses growth and development. Thus, the seed, although still hyper-responsive to ABA, is forced to germinate. (12) Even if ABA is not previously depleted, expansion growth, following a failure of the covering tissues, can slowly lead to break the development arrest and, then, to germination. Hence, (13) the seed’s covering tissues have an important role in restraining embryo expansion growth and, thus, preventing germination. Germination is often assessed in terms of the embryo (either coleoptile or radicle) growing through the seed covering tissues, and, therefore, to be detected it requires some growth, but this is (14) post-germinative seedling growth, which (15) is inhibited by extracellular ABA. (16) ABA and GAs down-regulate each other. They also have opposing effects, as GAs promote expansion growth (17), ABA-insensitive processes that lead to testa rupture (22) and (18) starch hydrolysis. The latter, in turn, (19) supports seedling growth. In case of wounding, ROS and NO· act together as an emergency signal breaking (20) the development arrest and (21) relieving ABA inhibition of growth. Modified and expanded after [61].

**Figure 3 plants-12-03963-f003:**
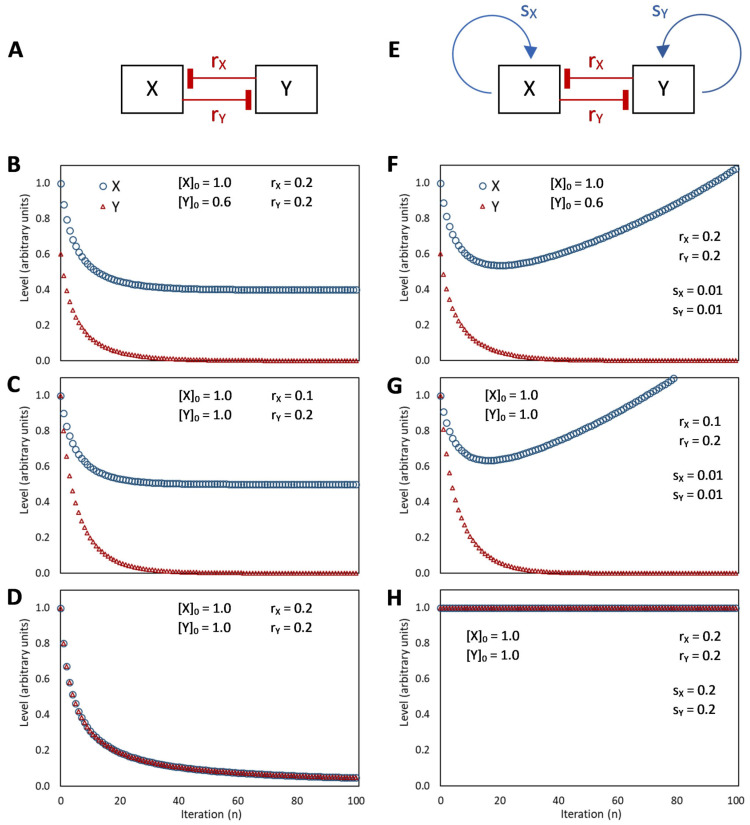
Basic regulatory circuits. (**A**) A bistable circuit consists of two regulative elements, X and Y, that negatively affect each other. The time evolution of the circuit is simulated as an iterative process, wherein the state of the circuit at each iteration depends on the levels of X and Y established at the previous iteration (apart from the initial state, for which they are given as [X]_0_ and [Y]_0_) and from the intensities of the repressive effects (given by the level of the repressing element times a given repression coefficient, that is, r_X_·[Y] and r_Y_·[X] for X and Y, respectively; which are also assumed to be proportional to the level of the repressed element, so that the latter is reduced percentually rather than linearly). Either one or the other element becomes predominant when (**B**) the initial level of such element is higher than the other, or (**C**), even if they are equal, the repression coefficient for that element is smaller. As only repressive effects exist, even the level of the element that ultimately prevails is reduced with respect to the initial state, and (**D**) if both the initial levels and the repression coefficients are equal, the whole circuit gradually dwindles away to nothing. (**E**) A so-called tristable circuit is obtained when self-enhancing effects are also present (given by the level of each element times a self-enhancing coefficient, that is, s_X_·[X] and s_Y_·[Y] for X and Y, respectively). This circuit amplifies the promotive effects on the initially favoured element, due to either (**F**) a higher initial level, or (**G**) a smaller repression coefficient (net of the self-enhancing effect). However, (**H**) if the intensity of repression matches the intensity of self-enhancement for each element, then no net change in the levels of the two regulative elements takes place, and the circuit attains an additional intermediate state that, even though highly unstable, can indefinitely retain the initial levels of both X and Y. The latter can be envisaged as the seed levels of GAs and ABA, respectively. A bistable circuit can, thus, be used to represent the antagonist interplay of ABA and GAs in the seeds according to the revised hormone-balance theory.

**Figure 4 plants-12-03963-f004:**
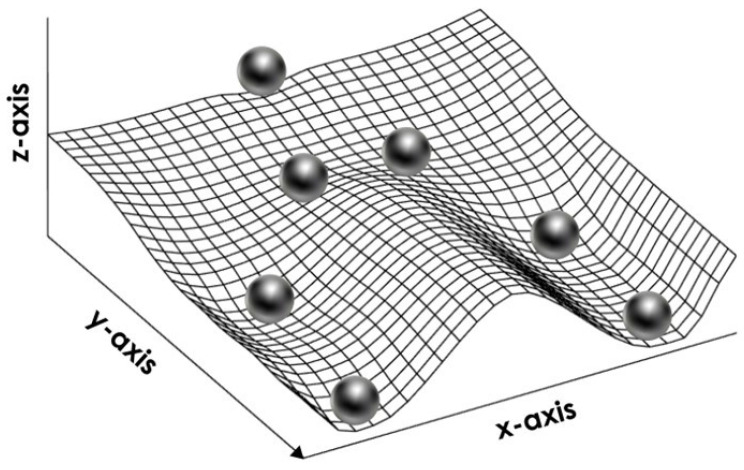
Development landscape for a system with a dichotomous outcome: the time-course is along the *y*-axis; events differentiate on the *x*-axis; and the *z*-axis (the vertical axis) represents improbability (that is, stability, inverted). Thus, valleys and basins correspond to more stable states, whereas peaks and ridges correspond to highly instable states. Accordingly, main changes are expected to occur downhill: from lower stability toward higher stability. A trajectory can be envisioned as the most probable path of a rolling ball, or the flux of a fluid, subject to downward attraction, like in a gravitational field. In the case of stability, notwithstanding the field being fictional, lower states are sometimes called ‘attractors’, though it is rather the greater instability of higher states that favours the shift in the regulatory network to a lower state. The landscape is intended to represent a self-contained part of the development: from an initial stage wherein the starting state of the biological system is reckoned as an input, to an end stage with a number (≥1) of possible outputs (meant as open valleys). The latter are considered irreversible once the system (represented as a metal marble here) reaches the front edge of the landscape. A development landscape can be used to represent the state of a seed along its development trajectory toward either germination or dormancy; for example, using the seed ABA/GAs balance as a proxy for the seed state.

**Figure 5 plants-12-03963-f005:**
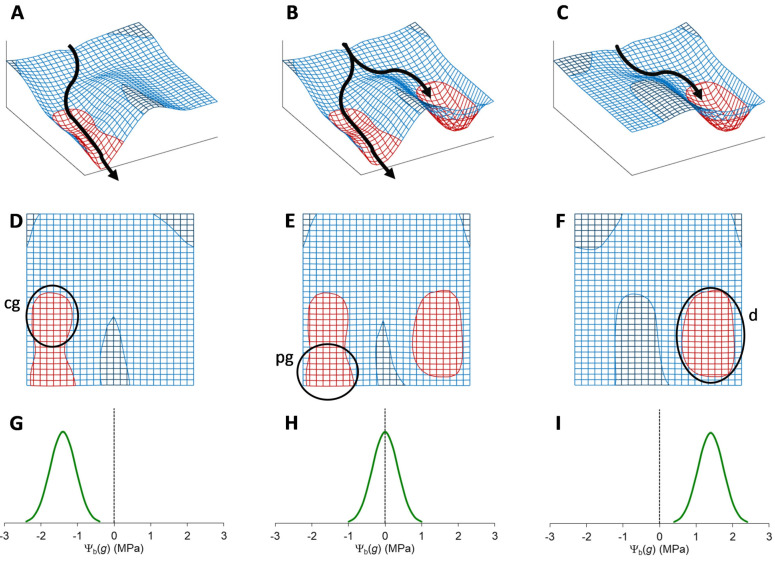
Conceptual representation of the binary development of living seeds imbibed in conditions favourable for germination. (**A**–**C**) Development landscapes of seed batches with different levels of dormancy. Black arrows represent the two possible developmental trajectories. (**D**–**F**) Contour graphs (2D projections) of the development landscapes. (**G**–**I**) Representations of the germinative potentials, across a seed sample, as a random distribution (the green bell-shaped curve) of base water-potential thresholds, according to the hydrotime model. The part of the seed population that is on the left of the vertical dashed line, which corresponds to 0 Mpa (that is, the water potential of pure water), germinate in water (as they have germinative potentials more negative than the incubation medium); those on the right do not (i.e., they are dormant). (**A**,**D**,**G**) Fully germinating seeds. (**B**,**E**,**H**) Partially dormant seeds. (**C**,**F**,**I**) Fully dormant seeds. Conditions that change the germinative status of a seed sample, like dry afterripening and stratification, modify the development landscape of development, thereby changing the relative probabilities of the two development trajectories. This corresponds to a shift in the random distribution of the seed population germinative potentials along the *x*-axis. Three notable states (marked by a red colour of the mesh) are evidenced (circled) in the contour graphs: (cg, commitment to germination) corresponds to the state of a seed that is going to germinate but has not yet attained physical modifications (i.e., no breaking of the seed coat has occurred); (pg) represents early post-germinative growth (that is, embryo expansion has caused the rupture of the seed coat); (d) is the seed dormant state (depicted as a basin, as it does not have a follow-through exit toward further development states). In favourable germination conditions, germinating seed smoothly pass through the first (cg) and the second (pg) state along the germinative trajectory; thus, they form a declining continuous red valley (leading to the natural development of the plant). Even though there is continuity between them for both fully germinating and partially dormant seeds, according to the hydrotime model, the more negative a seed base water-potential threshold is, the faster the seed passes from the first (cg) to the second (pg) state (i.e., the quicker germination is).

**Figure 6 plants-12-03963-f006:**
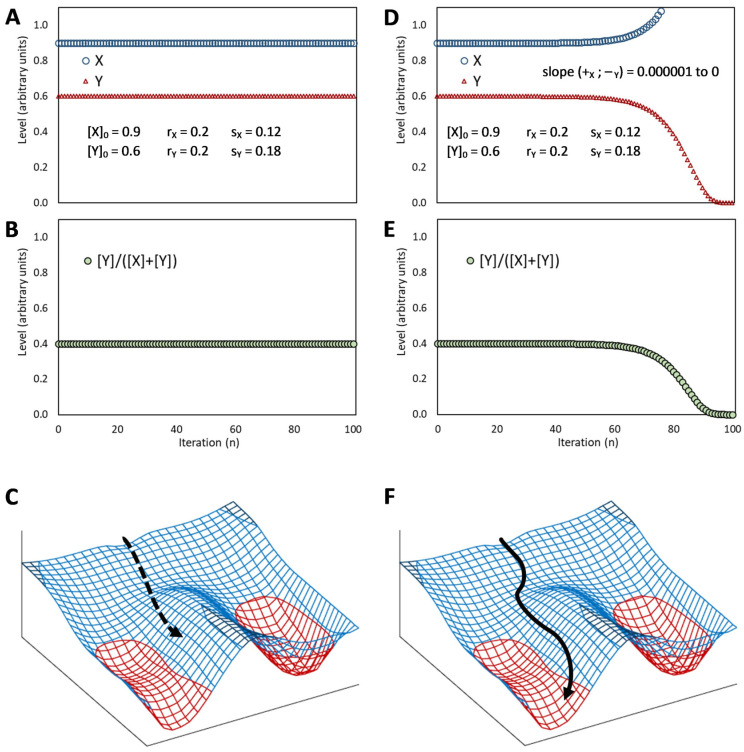
Relationship between a tristable circuit and the development landscape. (**A**) In a tristable circuit any combination of X and Y levels can be metastable provided that the repressive and self-inducing effects equate for each element, that is, r_X_·[Y]·[X] = s_X_·[X] and r_Y_·[X]·[Y] = s_Y_·[Y]. (**B**) The proportion of a given element to the sum of the two regulative elements can be assumed as a single indicator of the state of the circuit. (**C**) If the metastable circuit proceeds along a trajectory over a development landscape, the stability of the circuit is affected by the landscape. Specifically, as it moves transversally to a slope the dynamic equilibrium of the circuit is expected to eventually shift toward a lower state, changing the proportion of a given element in accordance with the equilibrium of the most stable trajectory of that side of the landscape. In this case, it is assumed that the valley corresponds to the predominance of the X element, whereas in the basin, Y predominates. (**D**) The action of the landscape may be modelled like a slope effect that, depending on the side of the landscape, affects one element positively (+slope_X_·[X]) and the other negatively (−slope_Y_·[Y]), and decreases as the circuit approaches the dynamic equilibrium of lowest state of that side, i.e., the corresponding most stable trajectory of the circuit. After a slow initial phase, the metastable state of the circuit is destabilized and the circuit then quickly shifts to either one or the other of the two stable states of the tristable circuit, depending on the slope. (**E**) The proportion of an element describes the state of the circuit: [Y]/([X] + [Y]), in this case. (**F**) The trajectory of a metastable state along the development landscape can be illustrated by the proportion of an element of the circuit to show that, though the metastable state can initially skip the most stable states at the early bottom of the valley (or basin), the slope of the landscape eventually drives that circuit to the corresponding most stable steady states. That is, the development landscape forces an individual circuit to follow its pre-defined development trajectory. Thus, as discussed in the main text, the proportion of ABA, that is, [ABA]/([GAs] + [ABA]), could be a better way to represent the seed state along its development trajectory with respect to the ABA/GAs ratio.

**Figure 7 plants-12-03963-f007:**
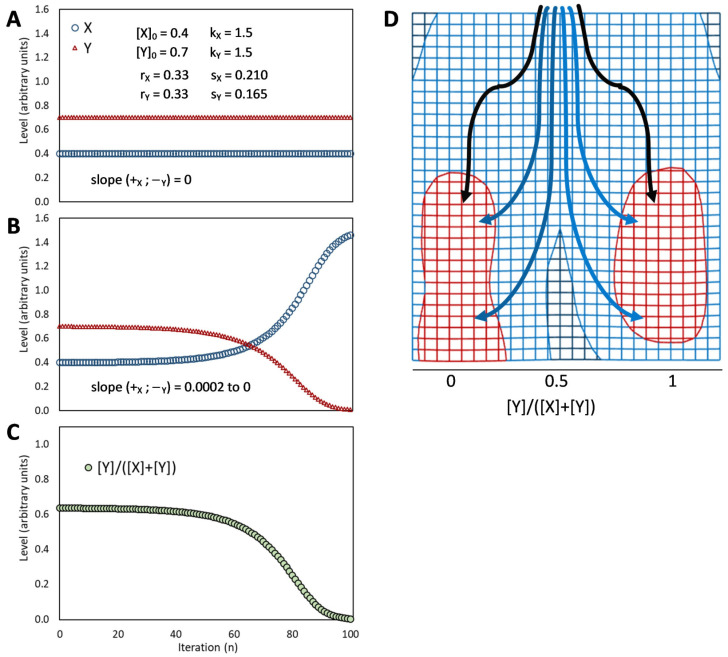
A self-limiting tristable circuit, in which a maximum stable level (k) is pre-established for each element, is suitable to illustrate a development trajectory along a development landscape. (**A**) Even in this tristable circuit, any combination of X and Y levels can be metastable provided that there is no slope effect, and the repressive and self-promoting effects equate for each element; in this case, r_X_·[Y]·[X] = s_X_·[X]·(k_X_ − [X]) and r_Y_·[X]·[Y] = s_Y_·[Y]·(k_Y_ − [Y]). (**B**) The presence of a slope effect shifts the circuit from the metastable state to the steady state most stable (the lowest) on the side (valley) of the landscape the circuit is moving through. (**C**) The state of the circuit is conveniently represented by the proportion of an element: [Y]/([X] + [Y]), in this case. (**D**) The individual paths of the seeds with different initial states (characterized by diverse initial proportions of a given element, shown on the *x*-axis below the graph) ultimately converge into either one or the other stable state of the tristable circuit (with either X or Y predominating). This happens even if the paths (blue arrows) of circuits that are in a metastable state in the middle between the two extreme stable steady states can initially deviate from the most probable development trajectories (black arrows), which always follow the lower states along either branch of the development landscape. States of the development landscape beyond the limit values of the *x*-axis are meant to have values of the predominating element that trespass the corresponding pre-established maximum (k), since each stable state is probabilistic in nature and it displays, therefore, random oscillations around the ideal value. The ultimate developmental fate of each individual circuit in the binary landscape is determined by the initial state of that circuit (which shows random variability around an average characteristic of the seed batch) and the shape of the developmental landscape. In a seed, it can be supposed that maximum physiological limits for GAs and ABA exist.

**Figure 8 plants-12-03963-f008:**
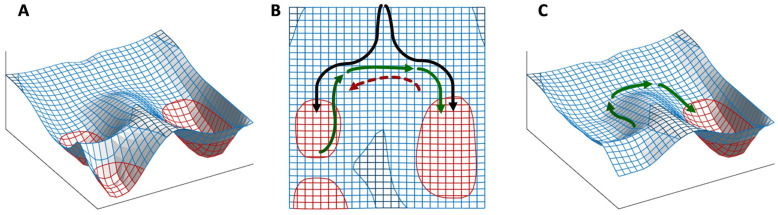
Upon imbibition, every living seed starts following either one development trajectory through the development landscape. (**A**) Under unfavourable conditions for germination, however, seeds whose regulatory network reaches a germinative state (i.e., commitment to germination) that is a prelude to germination, have not yet attained physical modifications, and they find their development trajectory toward germination precluded by a barrier in the developmental landscape brought about by the ambient conditions. The germination valley has, thus, become a basin. (**B**) On the contour graph, the binary development trajectories are displayed (black arrows). If the environmental conditions are suitable for dormancy induction (this can occur when conditions are unfavourable for germination), the regulatory networks of these seeds, like a blocked stream, can overflow into the dormancy basin (green arrows). With this inversion of flow, and exchange of basin, the seeds gradually acquire secondary dormancy. They may return to the germinative basin (dashed red arrow) if environment conditions quickly reverse to favour dormancy breaking and germination. However, (**C**) if environmental conditions suitable for dormancy induction persist, they gradually modify the development landscape by lifting the germinative basin. As this progresses, more and more non-dormant seeds are decanted into the dormancy basin.

## Data Availability

Data sharing is not applicable.

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
