# Peer review of "A Travel through Landscapes of Seed Dormancy"

_plants, 2023, doi:10.3390/plants12233963_

Round 1

Reviewer 1 Report

Comments and Suggestions for Authors

Seed dormancy is a programmatic developmental process. The acquisition of seed dormancy and the onset of germination are crucial for seed development and the life cycle of plants. This paper provides a comprehensive overview of the developmental fate of seeds, which deepens our understanding of many features of seed dormancy and germination. However, the manuscript, like a book chapter, has an awful length of 48 pages. It is difficult for readers to follow because some paragraphs (e.g., lines 1089-1180, 1874-1972) and figures (e.g., Figure 3 and Figure 4) that describe conceptual models of bistable and tristable regulatory circuits are not explicitly associated with the interplay of ABA and GA in the seeds. In addition, understanding the concept of partial dormancy and the metastable state of seed requires clarification at both the individual and population levels.

Author Response

Seed dormancy is a programmatic developmental process. The acquisition of seed dormancy and the onset of germination are crucial for seed development and the life cycle of plants. This paper provides a comprehensive overview of the developmental fate of seeds, which deepens our understanding of many features of seed dormancy and germination.

Thanks for appreciating my work.

However, the manuscript, like a book chapter, has an awful length of 48 pages. It is difficult for readers to follow because some paragraphs (e.g., lines 1089-1180, 1874-1972) and figures (e.g., Figure 3 and Figure 4) that describe conceptual models of bistable and tristable regulatory circuits are not explicitly associated with the interplay of ABA and GA in the seeds. In addition, understanding the concept of partial dormancy and the metastable state of seed requires clarification at both the individual and population levels.

Truly this manuscript ended up being much longer than I originally intended. The fact is that I tried to integrate different aspects of seed dormancy that, in my opinion, are better understood when looked at from a wide perspective. Notwithstanding “an awful length of 48 pages” some things were, indeed, not yet fully clear, as remarked by the Reviewer. Accordingly, I have tried to better clarify the points raised by the Reviewer in the revised manuscript. I have also made many cuts and several small revisions. Thanks for pointing out these weaknesses.

Reviewer 2 Report

Comments and Suggestions for Authors

This article has give a detailed systematic description of developmental landscape, very useful and meaningful in seed dormancy biology and seed quality management.

Just some issues:

It seems too long as one paper. It is better to concentrate it to 2/3 of the current size.

"This is somewhat analogous to the structure of a democratic country: the populace designates the government, thereafter, the government rules over the populace." in line 784 to 785 is not necessary and better to delete.

Comments on the Quality of English Language

English is generally fine. Only too many Parenthetical sentence pattern makes the article being look like an educational article than a scientific research one. I think the author may better chang some this sentence pattern into more science paper pattern.

Author Response

This article has give a detailed systematic description of developmental landscape, very useful and meaningful in seed dormancy biology and seed quality management.

Thank you for appreciating my work.

Just some issues:

It seems too long as one paper. It is better to concentrate it to 2/3 of the current size.

I must admit that this manuscript ended up being much longer than I originally intended. The fact is that I was trying to integrate different aspects of seed dormancy that, in my opinion, are better understood when looked at from a comprehensive perspective. As I was writing down my view, more and more details appeared to be relevant to show that different aspects of seed dormancy, apparently unrelated, ultimately integrate into a unique vision. So, I’m afraid that making extensive cuts to reduce the size of the paper by one third would mean to remove the discussion of important aspects and pertinent references, thereby worsening the comprehensibility of the several concepts discussed in the text. In fact, as remarked by another Reviewer, a few things were not yet fully clear. One of the reasons I chose to submit this manuscript to Plants is, indeed, that there is explicitly “no restriction on the maximum length of the papers” (as from the journal’s Aims). However, I carefully revised the whole manuscript to simplify and condensate the explanations that made this review to appear like an “educational article”, as mentioned below. I also made many small cuts and several minor improvements.

"This is somewhat analogous to the structure of a democratic country: the populace designates the government, thereafter, the government rules over the populace." in line 784 to 785 is not necessary and better to delete.

Deleted.

English is generally fine. Only too many Parenthetical sentence pattern makes the article being look like an educational article than a scientific research one. I think the author may better chang some this sentence pattern into more science paper pattern.

I tried to introduce some aspects usually not thought of as associated with seed dormancy, thus, I wished to explain them more didactically. As it appears I went too far in this respect, I re-structured several sentences according to a “more science paper pattern”. Thank you for pointing out this aspect.

Reviewer 3 Report

Comments and Suggestions for Authors

The idea to present seed dormancy through simple thermodynamic models is quite novel, interesting and gains reader's attention. The text is well-organized and easily readable, supported by the figures with well-developed captions. The number of citations is sufficient and properly chosen.

Remark: Blue arrow is also the dotted line, which should be mentioned in the caption to Fig. 8B. Please note thet for some readers the difference between blue and green may be uneasy to recognize, especially when the background is the blue grid.

Author Response

The idea to present seed dormancy through simple thermodynamic models is quite novel, interesting and gains reader's attention. The text is well-organized and easily readable, supported by the figures with well-developed captions. The number of citations is sufficient and properly chosen.

I’m happy you consider my review quite novel and interesting.

Remark: Blue arrow is also the dotted line, which should be mentioned in the caption to Fig. 8B. Please note thet for some readers the difference between blue and green may be uneasy to recognize, especially when the background is the blue grid.

Right, I used a dashed line to better distinguish the blue arrow, but I forgot to mention this in the figure caption. Thanks for pointing out this omission. I also changed its color to red to make it more evident.

Round 2

Reviewer 1 Report

Comments and Suggestions for Authors

I have no further comments on the revision. It is still too long for readers.

Author Response

I have no further comments on the revision. It is still too long for readers.

This is because in this review I tried to provide a critical update as well as an original perspective to look at seed dormancy. So, I must explain some specialistic concepts that are not often considered when dealing with seed dormancy and show that many different findings present in the literature can fit together when looked at from such perspective. Therefore, making extensive cuts would necessarily lead to reduce the immediate comprehensibility of some concepts and/or to exclude the discussion of pertinent references. Nevertheless, I made a few further cuts of sentences that were not strictly necessary for the main focus of the review. Although, on the one hand, I realize that many readers could find this review too long as a summary of the latest literature, on the other hand, making it shorter and thereby reducing the broadness of the view I intend to offer seems a losing tradeoff. The fact is that I never intended to just provide a quick summary of the literature.

I hope the Reviewer can accept my point of view, even if it goes against the mainstream opinion that a scientific review must always provide a concise update on the matter. In this respect, I understand that this review might seem more like an essay than a literature review, since essays “present provocative arguments aimed to stimulate the readers’ re-thinking of certain issues” (https://www.mdpi.com/about/article_types). Like for reviews, MDPI suggests a minimum length for essays, but not a maximum.

One of the reasons I chose to submit this manuscript to Plants is, indeed, that there is explicitly “no restriction on the maximum length of the papers” (as from the journal’s Aims). So, unless a Reviewer or the Academic Editor deem to point out specific parts of this Review that they regard as needlessly long and deserving to be reduced, or explanations and references that are not really useful, I believe that further cuts would worsen rather than improve the significance of this review.